# Movement of accessible plasma membrane cholesterol by the GRAMD1 lipid transfer protein complex

Tomoki Naito[1], Bilge Ercan[1], Logesvaran Krshnan[1†], Alexander Triebl[2], Dylan Hong Zheng Koh[1], Fan-Yan Wei[3], Kazuhito Tomizawa[3], Federico Tesio Torta[2], Markus R Wenk[2], Yasunori Saheki[1,4]*

[1]Lee Kong Chian School of Medicine, Nanyang Technological University, Singapore, Singapore; [2]Department of Biochemistry, Yong Loo Lin School of Medicine, National University of Singapore, Singapore, Singapore; [3]Department of Molecular Physiology, Faculty of Life Sciences, Kumamoto University, Kumamoto, Japan; [4]Institute of Resource Development and Analysis, Kumamoto University, Kumamoto, Japan

**Abstract** Cholesterol is a major structural component of the plasma membrane (PM). The majority of PM cholesterol forms complexes with other PM lipids, making it inaccessible for intracellular transport. Transition of PM cholesterol between accessible and inaccessible pools maintains cellular homeostasis, but how cells monitor the accessibility of PM cholesterol remains unclear. We show that endoplasmic reticulum (ER)-anchored lipid transfer proteins, the GRAMD1s, sense and transport accessible PM cholesterol to the ER. GRAMD1s bind to one another and populate ER-PM contacts by sensing a transient expansion of the accessible pool of PM cholesterol via their GRAM domains. They then facilitate the transport of this cholesterol via their StART-like domains. Cells that lack all three GRAMD1s exhibit striking expansion of the accessible pool of PM cholesterol as a result of less efficient PM to ER transport of accessible cholesterol. Thus, GRAMD1s facilitate the movement of accessible PM cholesterol to the ER in order to counteract an acute increase of PM cholesterol, thereby activating non-vesicular cholesterol transport.

*For correspondence:
yasunori.saheki@ntu.edu.sg

**Present address:** †Sir William Dunn School of Pathology, University of Oxford, Oxford, United kingdom

**Competing interests:** The authors declare that no competing interests exist.

## Introduction

Sterol is one of the major membrane lipids in eukaryotes. In metazoans, cholesterol represents ~20% of total cellular lipids and is therefore essential for the structural integrity of cellular membranes and for cell physiology (*van Meer et al., 2008*; *Vance, 2015*). Sterol is distributed among cellular membranes primarily via non-vesicular transport, a process that is independent of membrane traffic (*Baumann et al., 2005*; *Hao et al., 2002*; *Heino et al., 2000*; *Ikonen, 2008*; *Urbani and Simoni, 1990*). Levels of sterol vary considerably between different cellular membranes. Between 60% and 80% of total cellular cholesterol is concentrated in the plasma membrane (PM), where it represents up to ~45% of total lipids in this bilayer (*de Duve, 1971*; *Lange et al., 1989*; *Ray et al., 1969*). Cellular cholesterol levels are maintained by regulated delivery and production, primarily through receptor-mediated endocytosis of low-density lipoproteins (LDLs) (*Goldstein and Brown, 2015*) and de novo synthesis in the endoplasmic reticulum (ER) that is controlled by the activation of SREBP transcription factors (*Brown et al., 2018*; *Goldstein and Brown, 1990*). Cholesterol is also supplied to cells via high-density lipoproteins (HDL) through the reverse cholesterol flux pathway (*Acton et al., 1996*; *Phillips, 2014*).

Cholesterol within the bilayer membranes exists in two distinct chemical states: one being free and 'accessible' (also known as 'unsequestered' or 'chemically active'), and the other being

**eLife digest** The human body contains trillions of cells. At the outer edge of each cell is the plasma membrane, which protects the cell from the external environment. This membrane is mostly made of fatty molecules known as lipids and about half of these lipids are specifically cholesterol. Human cells can either take up cholesterol that were obtained via the diet or produce it within a compartment of the cell called the endoplasmic reticulum.

Cells need to monitor the cholesterol levels in both the endoplasmic reticulum and the plasma membrane in order to regulate the uptake or production of this lipid. For example, if there is too much of cholesterol in the plasma membrane, then the cell transports some to the endoplasmic reticulum to tell it to shut down cholesterol production. However, how these different areas of the cell communicate with each other, and transport cholesterol, has remained unclear.

Naito et al. set out to look for key regulators of cholesterol transport and identified a group of endoplasmic reticulum proteins called GRAMD1 proteins. Cholesterol in the plasma membrane is either accessible or inaccessible, meaning it either can or cannot be moved back into the cell. The GRAMD1 proteins sense accessible cholesterol, and experiments with human cells grown in the laboratory showed that, specifically, the GRAMD1 proteins work together in a complex to sense accessible cholesterol at or near the plasma membrane. One particular part of the protein senses when the amount of accessible cholesterol reaches a certain level at the plasma membrane; when this threshold is reached, the complex flips a switch to start the transport of cholesterol to the endoplasmic reticulum and tell it to shut down cholesterol production.

This coupling of sensing and transporting lipids by one protein complex also helps maintain the right ratio of accessible and inaccessible cholesterol in the plasma membrane to prevent cells from activating unwanted cell-signaling events. Getting rid of the GRAMD1 proteins in cells, or removing sensing part of these proteins, leads to inefficient transport of cholesterol. A better understanding of how GRAMD1 proteins sense the accessibility of cholesterol could potentially help identify new approaches to control cholesterol transport inside cells. This may in turn eventually lead to new treatments that counteract the defects in cholesterol metabolism seen in some forms of neurodegenerative diseases such as Alzheimer's disease and Parkinson's disease.

'inaccessible' (also known as 'sequestered' or 'chemically inactive') owing in part to the formation of complexes with other membrane lipids, including sphingomyelin and phospholipids (*Chakrabarti et al., 2017*; *Das et al., 2014*; *Gay et al., 2015*; *Lange et al., 2013*; *Lange et al., 2004*; *McConnell and Radhakrishnan, 2003*; *Ohvo-Rekilä et al., 2002*; *Radhakrishnan and McConnell, 2000*; *Sokolov and Radhakrishnan, 2010*). Most cholesterol in the PM is sequestered, but a small fraction of PM cholesterol (~15% of PM lipids) remains accessible for extraction and transport (*Das et al., 2014*). Although the majority of cellular cholesterol resides in the PM, the biosynthesis of cholesterol occurs exclusively in the ER. Thus, the ER must communicate with the PM to monitor levels of PM cholesterol and to adjust cholesterol biosynthesis in order to maintain lipid homeostasis. To achieve this, cells sense transient increases in the accessible pool of PM cholesterol and rapidly transport the newly expanded pool of accessible PM cholesterol to the ER. This suppresses cholesterol biosynthesis by inhibiting SREBP-2, a master regulator of de novo cholesterol synthesis, thereby avoiding cholesterol overaccumulation while maintaining PM cholesterol levels (*Das et al., 2014*; *Infante and Radhakrishnan, 2017*; *Lange and Steck, 1997*; *Lange et al., 2014*; *Scheek et al., 1997*; *Slotte and Bierman, 1988*). Artificially trapping the accessible pool of cholesterol in the PM results in dysregulated activation of SREBP-2 (*Infante and Radhakrishnan, 2017*; *Johnson et al., 2019*). Despite its critical importance, the intracellular transport machinery that senses the accessibility of PM cholesterol is unknown. This machinery is likely to respond to a sharp change in the accessibility of cholesterol on the cytoplasmic leaflet of the PM and to facilitate transport of accessible cholesterol from the PM to the ER, thereby helping the ER to communicate with the PM. Such a homeostatic system would also allow cells to monitor PM cholesterol accessibility in order to help to maintain cellular cholesterol homeostasis.

The ER extends throughout the cytoplasm, forming physical contacts with virtually all other cellular organelles and the PM (*Phillips and Voeltz, 2016*; *Wu et al., 2018*). Growing evidence indicates

that these membrane contact sites play critical roles in cellular physiology, including lipid exchange and delivery via non-vesicular lipid transport that is facilitated by lipid transfer proteins (LTPs) (*Antonny et al., 2018*; *Drin, 2014*; *Elbaz and Schuldiner, 2011*; *Holthuis and Menon, 2014*; *Jeyasimman and Saheki, 2019*; *Kumar et al., 2018*; *Lahiri et al., 2015*; *Lev, 2012*; *Luo et al., 2019*; *Nishimura and Stefan, 2019*; *Petrungaro and Kornmann, 2019*; *Saheki et al., 2016*; *Saheki and De Camilli, 2017a*; *Saheki and De Camilli, 2017b*; *Wong et al., 2018*). Thus, LTPs may participate in intracellular cholesterol transport and may help to maintain PM cholesterol homeostasis by regulating non-vesicular cholesterol transport between the PM and the ER at ER–PM contact sites.

Decades of biochemical and genetic research into cholesterol metabolism has identified several key LTPs that bind to cholesterol and mediate its non-vesicular transport (*Luo et al., 2019*; *Wong et al., 2018*). These proteins include a family of 15 proteins that contain a StAR-related lipid transfer (StART) domain, which binds and transports a wide variety of lipids, including cholesterol, glycerolipids, and sphingolipids (*Alpy and Tomasetto, 2014*). Five members of this family, namely STARD1, STARD3, STARD4, STARD5, and STARD6, bind and transport cholesterol (*Alpy et al., 2013*; *Iaea et al., 2017*; *Lin et al., 1995*; *Mesmin et al., 2011*; *Soccio et al., 2002*; *Stocco, 2001*; *Wilhelm et al., 2017*), but they are not conserved in yeast. This lack of conservation suggests that there may be a more ancient family of sterol transfer proteins that control cholesterol homeostasis in all eukaryotes.

A bioinformatics search for proteins that possess StART-like domains identified a novel family of evolutionarily conserved proteins that includes six Lam/Ltc proteins in budding yeast (*Gatta et al., 2015*; *Murley et al., 2015*), and five GRAM domain-containing proteins (GRAMDs) in metazoans. These GRAMDs include the StART-like domain-containing GRAMD1s, also known as Asters (GRAMD1a/Aster-A, GRAMD1b/Aster-B, and GRAMD1c/Aster-C), and two highly related proteins that lack a StART-like domain (GRAMD2 and GRAMD3). Lam/Ltc proteins and GRAMDs all possess an N-terminal GRAM domain, which has structural similarity to the PH domain and thus may sense or bind lipids (*Begley et al., 2003*; *Tong et al., 2018*), and a C-terminal transmembrane domain, which anchors the proteins to the ER. Structural and biochemical studies of yeast and mammalian StART-like domains have identified a hydrophobic cavity that can bind sterol (*Gatta et al., 2018*; *Horenkamp et al., 2018*; *Jentsch et al., 2018*; *Sandhu et al., 2018*; *Tong et al., 2018*). The StART-like domains of GRAMD1s bind and transport sterols in vitro (*Horenkamp et al., 2018*; *Sandhu et al., 2018*). Recent studies have demonstrated that some GRAMDs, including GRAMD1a, GRAMD1b, and GRAMD2, localize to ER–PM contact sites (*Besprozvannaya et al., 2018*; *Sandhu et al., 2018*). GRAMD2 facilitates STIM1 recruitment to ER–PM contacts and potentially regulates $Ca^{2+}$ homeostasis (*Besprozvannaya et al., 2018*), whereas GRAMD1b facilitates the transport of HDL-derived cholesterol to the ER in the adrenal glands of mice (*Sandhu et al., 2018*). By contrast, yeast Lam/Ltc proteins sense cellular stress and potentially regulate cholesterol exchange between the ER and other membranes (*Gatta et al., 2015*; *Murley et al., 2015*; *Murley et al., 2017*). However, the role of these proteins in PM cholesterol or sterol homeostasis has been elusive. In this study, we provide evidence that GRAMD1s sense a transient expansion of the accessible pool of PM cholesterol and facilitate its transport to the ER at ER–PM contact sites, thereby contributing to PM cholesterol homeostasis.

We found that GRAMDs form homo- and heteromeric complexes via their transmembrane domains and predicted the existence of luminal amphipathic helices that interact with each other within these complexes. We also found that GRAMD1s rapidly move to ER–PM contacts upon acute hydrolysis of sphingomyelin in the PM. We characterized the mechanisms of this acute recruitment and found that the GRAM domain acts as a coincidence detector of unsequestered/accessible cholesterol and anionic lipids in the PM, including phosphatidylserine, allowing the GRAMD1s to sense a transient expansion of the accessible pool of PM cholesterol once it increases above a certain threshold. We generated HeLa cells that lacked GRAMD1a/1b/1c (i.e., all of the GRAMDs that contain a StART-like domain) and determined the effect of knocking out these proteins on cholesterol metabolism, using a combination of cholesterol-sensing probes for live cell imaging and lipidomics of membrane extracts. Upon treatment with sphingomyelinase, which liberates the sphingomyelin-sequestered pool of PM cholesterol into the 'accessible' pool and thus stimulates its PM to ER transport, GRAMD1 triple knockout (TKO) cells exhibited exaggerated accumulation of the accessible pool of PM cholesterol and reduced suppression of SREBP-2 cleavage compared to wild-type

control cells. This accumulation resulted from less efficient transport of accessible cholesterol from the PM to the ER.

Using structure–function analysis, we demonstrated that GRAMD1s couple their PM-sensing property and cholesterol-transport function via their GRAM and StART-like domains, and that GRAMD1 complex formation ensures the progressive recruitment of GRAMD1 proteins to ER–PM contacts. Finally, we observed striking expansion of the accessible pool of PM cholesterol in GRAMD1 TKO cells at steady state. Drug-induced acute recruitment of GRAMD1b to ER–PM contacts was sufficient to facilitate removal of the expanded pool of accessible cholesterol from the PM in GRAMD1 TKO cells. Collectively, our findings provide evidence for novel cellular mechanisms by which GRAMD1s monitor and help to maintain PM cholesterol homeostasis in mammalian cells. As one of the key homeostatic regulators, GRAMD1s sense a transient expansion of the accessible pool of PM cholesterol and facilitate its transport to the ER, thereby contributing to PM cholesterol homeostasis at ER–PM contact sites.

## Results

### GRAMD proteins form homo- and heteromeric complexes

Previous studies identified GRAMD1s as ER-resident proteins that are distributed throughout ER structures in a punctate pattern (*Sandhu et al., 2018*). GRAMDs (namely GRAMD1a, GRAMD1b, GRAMD1c, GRAMD2, and GRAMD3) all possess an N-terminal GRAM domain and a C-terminal transmembrane domain. In addition, the three GRAMD1 proteins (GRAMD1s) possess a StART-like domain (*Figure 1A*). Some LTPs are known to form homo- and heteromeric complexes. Thus, we reasoned that GRAMD1s may also interact with one another to form complexes. To further analyze the dynamics of these proteins on the ER at high spatial resolution, we tagged the GRAMD1s, as well as GRAMD3, with fluorescent proteins and analyzed their localization using spinning disc confocal microscopy coupled with structured illumination (SDC-SIM). Analysis of COS-7 cells expressing individual EGFP-tagged GRAMD1s or GRAMD3 (EGFP-GRAMD1a, EGFP-GRAMD1b, EGFP-GRAMD1c, or EGFP-GRAMD3) and a general ER marker (RFP-tagged Sec61β) revealed enrichment of GRAMD1s and GRAMD3 in similar discrete patches along ER tubules. By contrast, RFP-Sec61β localized to all domains of the ER, including the nuclear envelope and the peripheral tubular ER network (*Hoyer et al., 2018*) (*Figure 1B* and *Figure 1—figure supplement 1A*). When individual EGFP–GRAMD1s and either mRuby-tagged GRAMD1b (mRuby-GRAMD1b) (*Figure 1C*) or mCherry-tagged GRAMD3 (mCherry-GRAMD3) (*Figure 1—figure supplement 1B*) were co-expressed in COS-7 cells, the patches of EGFP and mRuby/mCherry significantly overlapped, indicating potential complex formation between these proteins on tubular ER.

To test whether these proteins form complexes, we examined biochemical interactions between GRAMD1s and GRAMD3 using co-immunoprecipitation assays. HeLa cells co-transfected with individual EGFP–GRAMD1s together with either myc-tagged GRAMD1b (Myc–GRAMD1b) (*Figure 1D* and *Figure 1—figure supplement 1C*) or myc-tagged GRAMD3 (Myc–GRAMD3) (*Figure 1E* and *Figure 1—figure supplement 1D*) were lysed, and either anti-GFP (*Figure 1D,E*) or anti-Myc nanobodies (*Figure 1—figure supplement 1C,D*) were used to perform immunoprecipitation. Analysis of the resulting immunoprecipitates by western blotting (i.e. immunoblotting) revealed robust interaction between GRAMD1s and GRAMD1b (*Figure 1D* and *Figure 1—figure supplement 1C*), as well as between GRAMD1s and GRAMD3 (*Figure 1E* and *Figure 1—figure supplement 1D*). These results demonstrate that these proteins form both homo- and heteromeric complexes.

### Luminal helices and transmembrane domains of GRAMD proteins are important for their complex formation

The formation of homo- and heteromeric complexes between GRAMD1s and GRAMD3 suggested the presence of amino-acid sequence within these proteins that facilitate their interaction. Secondary structure predictions indicated the presence of a conserved alpha helix within the luminal region of GRAMD1s (*Figure 2A*). Furthermore, helical wheel analysis of the luminal helix from GRAMD1b predicted that this protein contained an amphipathic helix, with charged and hydrophobic amino acids occupying opposite sides of the helix (*Figure 2B* and *Figure 2—figure supplement 1A,B*). It is known that some amphipathic helices mediate protein–protein interactions through their

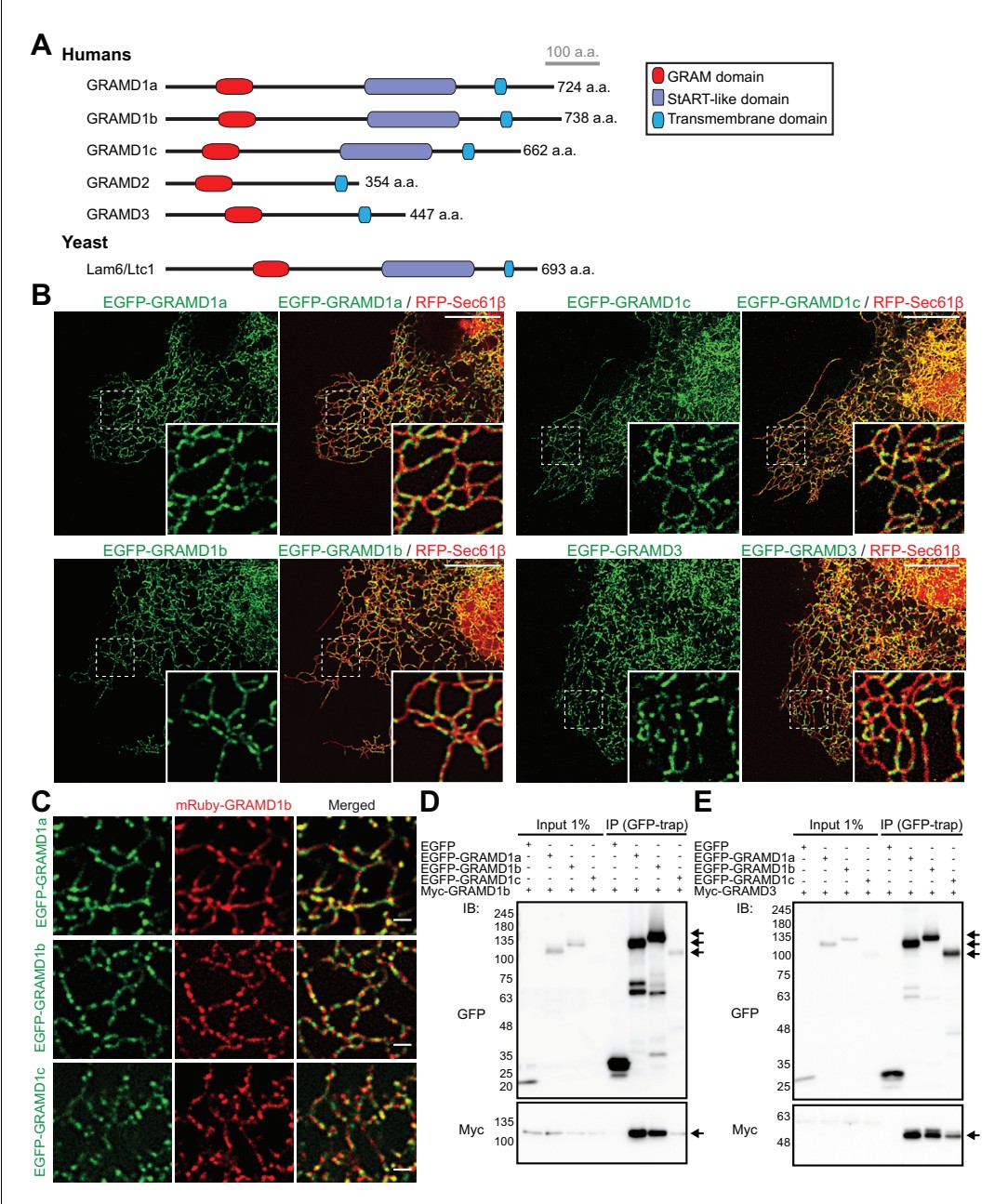

**Figure 1.** GRAMD proteins form homo- and heteromeric complexes. (**A**) Domain structure of GRAMD proteins in comparison to yeast Lam6/Ltc1. (**B**) Confocal images of live COS-7 cells expressing the ER membrane marker RFP-Sec61β and EGFP–GRAMD protein constructs as indicated. Insets show at higher magnification the regions indicated by white dashed boxes. Note the presence of the patches of EGFP–GRAMDs throughout the tubular ER. Scale bars, 10 μm. (**C**) Confocal images of live COS-7 cells expressing mRuby-GRAMD1b and EGFP–GRAMD1s as indicated. Note the presence of mRuby–GRAMD1b patches that partially overlap with the patches of EGFP–GRAMD1s. Scale bars, 1 μm. (**D, E**) Extracts of HeLa cells transfected with the indicated constructs were subjected to anti-GFP immunoprecipitation (IP) and then processed for SDS-PAGE and immunoblotting (IB) with anti-GFP and anti-Myc antibodies. Inputs are 1% of the total cell lysates. Note the strong biochemical interaction between GRAMD1b and GRAMD1s (**D**) and between GRAMD3 and GRAMD1s (**E**). Immunoprecipitated EGFP-GRAMD1s, Myc-GRAMD1b and Myc-GRAMD3 are indicated by arrows.

The online version of this article includes the following figure supplement(s) for figure 1:

**Figure supplement 1.** GRAMD proteins form homo- and heteromeric complexes.

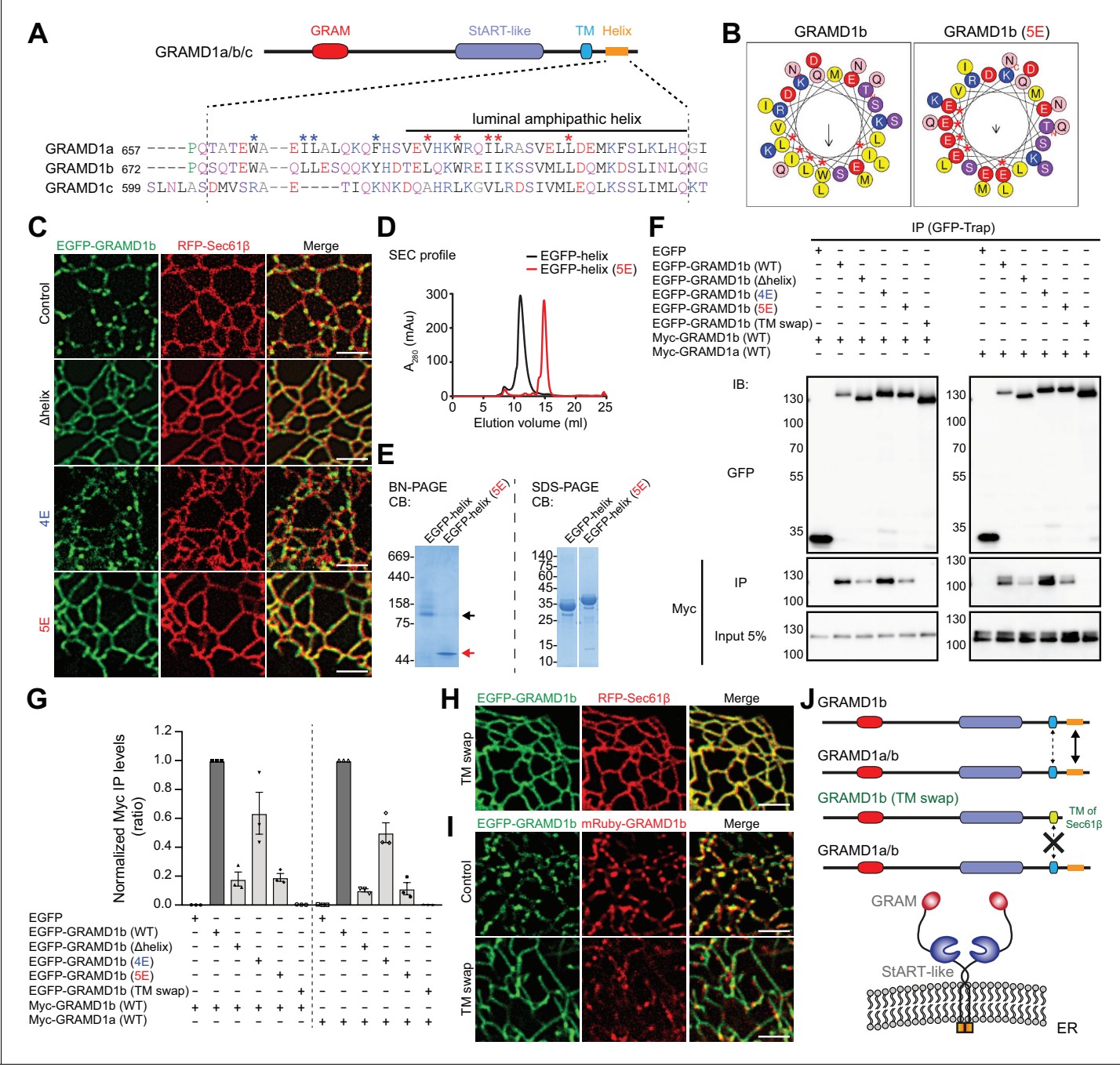

**Figure 2.** Luminal helix and transmembrane domain of GRAMD1b are important for homo- and heteromeric interaction. (**A**) Sequence alignment of the luminal region of GRAMD1s. This region is predicted by Phyre2 to contain an amphipathic helix (*Kelley et al., 2015*) as indicated. Blue and red asterisks mark hydrophobic amino acid residues that are partially conserved in GRAMD1s. The shared identities of the amino acid sequences of the amphipathic helices predicted by BLAST analysis were: 75% (GRAMD1a vs. GRAMD1b); 75% (GRAMD1a vs. GRAMD1c); and 80% (GRAMD1b vs. GRAMD1c). The effects of the mutations of these residues to glutamic acid (4E in the case of blue marks; 5E in the case of red marks) were tested in GRAMD1b. Black, red, blue, and pink/purple colors denote hydrophobic, acidic, basic, and hydrophilic amino acid residues, respectively. (**B**) Predicted luminal amphipathic helix region of wild-type GRAMD1b (left panel) and that with L693E, W696E, I699E, I700E and L707E (5E) mutations (right panel) are shown as helical wheel representations. Predictions were made with the Heliquest server (*Gautier et al., 2008*). (**C**) Confocal images of live COS-7 cells expressing RFP–Sec61β and EGFP fusions of various GRAMD1b constructs [control, wild-type GRAMD1b; Δhelix, GRAMD1b lacking the predicted luminal amphipathic helix; 4E, GRAMD1b with 4E mutations in the luminal region (W678E, L681E, L682E, Y688E); 5E, GRAMD1b with 5E mutations in the predicted luminal amphipathic helix]. Note the reduced formation of GRAMD1b patches in Δhelix and 5E mutants but not in the 4E mutant. Scale bars, 2 μm. (**D**) Overlay of the size exclusion chromatography (SEC) profiles of the recombinant EGFP-tagged luminal helix region of wild-type

*Figure 2 continued on next page*

*Figure 2 continued*

GRAMD1b (EGFP–helix) and EGFP–helix with the 5E mutations [EGFP–helix (5E)]. Note the difference in elution volumes, indicating the formation of complexes mediated by the wild-type luminal helix. (**E**) Blue native (BN)-PAGE analysis (left panel) and SDS-PAGE analysis (right panel) of SEC-purified EGFP–helix and EGFP–helix (5E). Black and red arrows indicate the major bands for EGFP–helix and EGFP–helix 5E, respectively. Note the difference in their migration pattern in BN-PAGE. CB, Colloidal blue staining. (**F**) Extracts of HeLa cells transfected with the constructs as indicated were subjected to anti-GFP immunoprecipitation (IP) and then processed for SDS-PAGE and immunoblotting (IB) with anti-GFP and anti-Myc antibodies. Inputs are 5% of the total cell lysates. Note that the interaction of GRAMD1b or GRAMD1a is much reduced in GRAMD1b Δhelix or 5E mutants and abolished in the GRAMD1b (TM swap) mutant (GRAMD1b with its transmembrane domain and luminal region replaced with those of Sec61β) when compared to the levels of interactions seen in cells with wild-type GRAMD1b. This reduction is smaller in the GRAMD1b 4E mutant. (**G**) Quantification of the co-immunoprecipitation experiments shown in (**F**). The ratio of the band intensity of the co-immunoprecipitated Myc–GRAMD1b (left) or Myc–GRAMD1a (right) over that of the indicated immunoprecipitated EGFP-tagged proteins were calculated. The values were then normalized by the ratio of the band intensity of Myc–GRAMD1b over that of EGFP–GRAMD1b (WT) (left) or by the ratio of the band intensity of Myc–GRAMD1a over that of EGFP–GRAMD1b (WT) (right) [mean ± SEM, n = 3 IPs for each sample]. (**H**) Confocal images of a live COS-7 cell expressing RFP-Sec61β and EGFP-tagged GRAMD1b (TM swap). Scale bars, 2 μm. (**I**) Confocal images of live COS-7 cells expressing mRuby–GRAMD1b and EGFP fusions of GRAMD1b constructs [Control, wild-type GRAMD1b; TM swap, GRAMD1b (TM swap)]. Note the abolished formation of GRAMD1b patches in TM swap mutants. Scale bars, 2 μm. (**J**) Model of the homo- and heteromeric interactions of GRAMD1a/b. Their complex formation is facilitated primarily by their luminal amphipathic helices and additionally mediated by their transmembrane domains. These regions are important for the ability of GRAMD1s to form complexes and patches on the tubular ER network.

The online version of this article includes the following source data and figure supplement(s) for figure 2:

**Source data 1.** Dataset for *Figure 2*.
**Figure supplement 1.** Luminal helix of GRAMD1b is important for homo- and heteromeric interaction.

hydrophobic surfaces (*Segrest et al., 1990*). Therefore, we first asked whether the luminal helix was necessary for these proteins to form discrete patches on tubular ER. We focused on GRAMD1b as a model protein for analysis of the properties of the GRAMD1 luminal helices, generating a version of GRAMD1b that lacked the luminal helix (Δhelix), and a second version in which the five hydrophobic residues within the luminal helix were mutated to glutamic acid (5E), thereby disrupting the hydrophobic surface (*Figure 2B* and *Figure 2—figure supplement 1C*). Whereas GRAMD1b (wild-type control) formed patches on tubular ER, both GRAMD1b (Δhelix) and GRAMD1b (5E) exhibited diffuse localization patterns, with fewer discrete patches on tubular ER (*Figure 2C*). By contrast, a version of GRAMD1b in which the four hydrophobic residues preceding the luminal helix were mutated to glutamic acid (4E) formed patches that were similar to those formed by the control (*Figure 2C*), demonstrating that the 5E mutation specifically disrupted patch formation.

The potential ability of the luminal helices to interact directly with one another was examined using cell-free assays. Wild-type luminal helices (GRAMD1b$_{674–718}$) and luminal helices with the 5E mutation (GRAMD1b$_{674–718\ 5E}$) were purified individually as EGFP fusion proteins and analyzed by size exclusion chromatography (SEC). Whereas the predicted molecular weights of the fusion proteins were the same (~35 kDa), wild-type luminal helices (EGFP–helix: EGFP–GRAMD1b$_{674–718}$) eluted at a much lower elution volume compared to 5E mutant luminal helices [EGFP–helix (5E): EGFP–GRAMD1b$_{674–718\ 5E}$] (*Figure 2D*). Blue native PAGE analysis (BN-PAGE) of the purified proteins revealed that wild-type helices migrated slower than the 5E mutants, indicating that interaction between luminal helices depended on the hydrophobic surface of GRAMD1b (*Figure 2E*). By contrast, in the presence of SDS, the denatured forms of these proteins migrated similarly (SDS-PAGE). Slightly slower migration of 5E mutants on the gel was possibly due to the increased hydrophilicity of this fragment compared to wild-type (*Guan et al., 2015*) (*Figure 2E*). These results suggest that the luminal helix is probably amphipathic and is important for the formation of GRAMD1b complexes through its hydrophobic surface.

Finally, the formation of GRAMD1 complexes was examined biochemically in cells using co-immunoprecipitation assays. Homomeric interactions between GRAMD1bs and heteromeric interactions between GRAMD1b and GRAMD1a were greatly reduced when the luminal helix of GRAMD1b was either removed (Δhelix) or mutated to the 5E version, supporting the important role of the luminal helix in homo- and heteromeric interactions of the GRAMD1s (*Figure 2F,G*). Residual interactions were mediated by the transmembrane domain of GRAMD1b, as replacing this domain and its luminal region with those from Sec61β (TM swap) (*Figure 2J*) completely abolished the ability of GRAMD1b to form homo- and heteromeric complexes (*Figure 2F,G*). Accordingly, GRAMD1b with

the TM swap exhibited a diffuse localization pattern compared to that of wild-type GRAMD1b (*Figure 2H*), and failed to interact with wild-type GRAMD1b on tubular ER (*Figure 2I*). Thus, both transmembrane domains and luminal helices contributed to the formation of GRAMD1 complexes (*Figure 2J*). Taken together, these results revealed the biochemical mechanisms by which GRAMDs form homo- and heteromeric complexes. As key residues contributing to the hydrophobic surface of the luminal helix are conserved among GRAMD1s (*Figure 2A* and *Figure 2—figure supplement 1A*), they probably play a role in the heteromeric interactions of all of these proteins.

## The GRAM domain of GRAMD1s acts as a coincidence detector of unsequestered/accessible cholesterol and anionic lipids, and senses the accessibility of cholesterol

Recent studies demonstrated that 'cholesterol loading' leads to the accumulation of GRAMD1s at ER–PM contact sites (*Sandhu et al., 2018*). Within 20 min of treating cells with a complex of cholesterol and methyl-β-cyclodextrin (cholesterol/MCD), GRAMD1b was indeed recruited to the PM (*Figure 3A,B*; *Video 1*). In addition, we found that GRAMD1a, GRAMD1c, and GRAMD3 were all recruited to ER–PM contacts upon cholesterol loading, with kinetics similar to GRAMD1b recruitment (*Figure 3B*). However, a version of GRAMD1b that lacked the GRAM domain (GRAMD1b ΔGRAM) failed to localize to the PM, even after 30 min, indicating the essential role of this domain in sensing PM cholesterol (*Figure 3—figure supplement 1A*; *Video 2*). Although these results suggest that PM cholesterol plays a critical role in recruiting GRAMDs to ER–PM contacts, all of the GRAMDs localize to tubular ER at rest, even though a significant amount of cholesterol is already present in the PM (*Lange et al., 1989*; *Ray et al., 1969*). Thus, their GRAM domains may possess unique abilities to sense the accessibility of PM cholesterol, rather than detecting the total levels of PM cholesterol. However, it is not known whether the GRAM domains are able to sense accessible cholesterol in the PM.

To elucidate the biochemical properties of the GRAMD1 GRAM domain, we first purified the GRAM domain of GRAMD1b and performed liposome sedimentation assays to test its ability to bind lipids. In this assay, purified GRAM domains were mixed with sucrose-loaded heavy liposomes in sucrose-free buffer. After incubation, free liposomes and the liposomes that bound to GRAM domains (P) were pelleted by centrifugation; the supernatant contained only unbound GRAM domains (S) (*Figure 3C,E*, *Figure 3—figure supplement 1B,D*, and *Figure 3—figure supplement 2A,B*). The GRAM domain did not bind liposomes when the liposome contained only phosphatidylcholine (*Figure 3C* and *Figure 3—figure supplement 1B*). By contrast, the GRAM domain bound liposomes that contained free cholesterol, although such binding was rather weak, and only ~25% of purified GRAM domains bound liposomes even when the liposome contained high levels of cholesterol (Chol) (60%) (*Figure 3—figure supplement 1B*). The GRAM domain also bound liposomes when the liposomes contained phosphatidylserine (PS), the predominant anionic phospholipid in the PM. However, such binding only occurred when the liposomes contained non-physiological high levels of phosphatidylserine (50% or 80%) (*Figure 3—figure supplement 1B*). Thus, we explored the possibility that the GRAM domain may bind to membranes more efficiently in the presence of both lipids, thereby acting as a coincidence detector of unsequestered/accessible cholesterol and phosphatidylserine.

Little binding was observed when the liposomes contained 50% cholesterol or 20% phosphatidylserine (*Figure 3C* and *Figure 3—figure supplement 1B*). However, strong binding was observed when 50% cholesterol and 20% phosphatidylserine were both present in the liposomes (~80% of the GRAM domains bound to liposomes) (*Figure 3C*). Thus, the addition of free cholesterol dramatically enhanced binding of the GRAM domain to phosphatidylserine-containing membranes. Replacing cholesterol with a non-bilayer forming lipid, phosphatidylethanolamine (PE), abolished the binding of the GRAM domains to liposomes, confirming the specific effect of cholesterol (*Figure 3—figure supplement 2A*). Similar synergistic effects were observed with the GRAM domain of GRAMD1a (*Figure 3D*), suggesting the conserved function of GRAMD1 GRAM domains.

Despite the presence of phosphatidylserine (~10% of PM lipids) and high levels of cholesterol (~45% of PM lipids) in the PM of mammalian cells, GRAMD1s are not enriched at ER–PM contacts at rest (*Figure 1B*, *Figure 3A* and *Figure 4—figure supplement 3B*). The majority of cholesterol in the PM (~27% of PM lipids) is sequestered and 'inaccessible' to cytosolic proteins, and only ~15% of PM lipids remain unsequestered and accessible (*Das et al., 2014*). Thus, interactions between GRAM

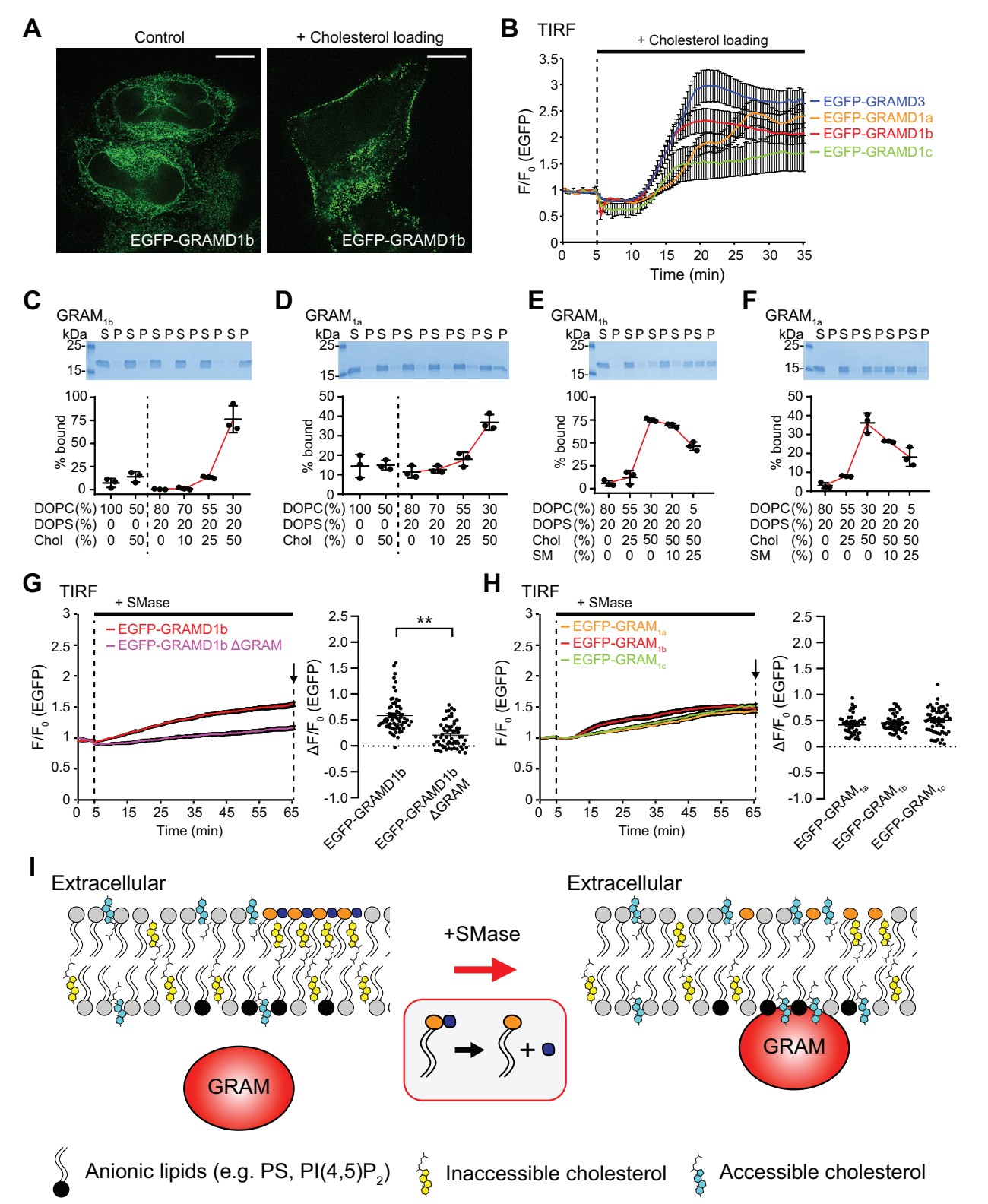

**Figure 3.** The GRAM domain of GRAMD1s acts as a coincidence detector of unsequestered/accessible cholesterol and anionic lipids, and senses a transient expansion of the accessible pool of cholesterol in the PM. (**A**) Confocal images of live HeLa cells expressing EGFP–GRAMD1b with or without cholesterol loading [the treatment with cholesterol/MCD complex (200 µM) for 30 min at 37°C]. Note the extensive recruitment of GRAMD1b to the PM upon cholesterol loading. Scale bars, 10 µm. (**B**) Time course of normalized EGFP signal, as assessed by total internal reflection fluorescence (TIRF)

*Figure 3 continued on next page*

*Figure 3 continued*

microscopy, from HeLa cells expressing EGFP–GRAMD protein constructs as indicated. Cholesterol loading [the treatment with cholesterol/MCD complex (200 µM)] is indicated. [mean ± SEM, n = 24 cells (EGFP–GRAMD1a), n = 29 cells (EGFP–GRAMD1b), n = 25 cells (EGFP–GRAMD1c), n = 28 cells (EGFP–GRAMD3; data are pooled from one experiment for GRAMD1a and two experiments for GRAMD1b, GRAMD1c and GRAMD3.] (C–F). Liposome sedimentation assays of the GRAM domain of GRAMD1b (GRAM$_{1b}$) and GRAMD1a (GRAM$_{1a}$). Liposomes containing the indicated mole% lipids were incubated with purified GRAM$_{1b}$ proteins (C, E) or purified GRAM$_{1a}$ proteins (D, F). Bound proteins [pellet, (P)] were separated from the unbound proteins [supernatant, (S)], run on SDS-PAGE and visualized by colloidal blue staining (mean ± SEM, n = 3 independent experiments for all the conditions). DOPC, phosphatidylcholine (1,2-dioleoyl-sn-glycero-3-phosphocholine); DOPS, phosphatidylserine (1,2-dioleoyl-sn-glycero-3-phospho-L-serine); Chol, cholesterol; SM, sphingomyelin (N-oleoyl-D-erythro-sphingosylphosphorylcholine). (G) Left: time course of normalized EGFP signal in response to sphingomyelinase (SMase), as assessed by TIRF microscopy of HeLa cells expressing EGFP–GRAMD1b or EGFP–GRAMD1b ΔGRAM. The treatment with SMase (100 mU/ml) is indicated. Right: values of ΔF/F$_0$ corresponding to the end of the experiment as indicated by the arrow [mean ± SEM, n = 72 cells (EGFP–GRAMD1b), n = 64 cells (EGFP–GRAMD1b ΔGRAM); data are pooled from three independent experiments for each condition; two-tailed unpaired Student's t-test, **p<0.0001]. (H) Left: time course of normalized EGFP signal in response to SMase, as assessed by TIRF microscopy of HeLa cells expressing the indicated EGFP-tagged GRAM domain of GRAMD1s. The treatment with SMase (100 mU/ml) is indicated. Right: values of ΔF/F$_0$ corresponding to the end of the experiment as indicated by the arrow [mean ± SEM, n = 48 cells (EGFP–GRAM$_{1a}$), n = 50 cells (EGFP–GRAM$_{1b}$), n = 58 cells (EGFP–GRAM$_{1c}$); data are pooled from two to three independent experiments for each condition]. (I) Schematics showing the interaction of the GRAM domain of GRAMD1s with the plasma membrane (PM) before and after sphingomyelinase (SMase) treatment. Left: at rest, subthreshold levels of accessible cholesterol in the PM are not sufficient to induce interaction of the GRAM domain with the PM. Right: liberation of the sphingomyelin-sequestered pool of cholesterol by SMase treatment leads to an increase in accessible cholesterol in the PM beyond the threshold, and induces PM recruitment of the GRAM domain as it senses both increase in unsequestered/accessible cholesterol and the presence of anionic lipids in the PM.

The online version of this article includes the following source data and figure supplement(s) for figure 3:

**Source data 1.** Dataset for *Figure 3*.
**Figure supplement 1.** The GRAM domain of GRAMD1s acts as a coincidence detector of unsequestered/accessible cholesterol and anionic lipids.
**Figure supplement 1—source data 1.** Dataset for *Figure 3—figure supplement 1*.
**Figure supplement 2.** The GRAM domain of GRAMD1s binds to membranes by sensing cholesterol accessibility.
**Figure supplement 2—source data 1.** Dataset for *Figure 3—figure supplement 2*.

domains and the PM could be suppressed by 'the factors that sequester cholesterol' in this bilayer in cells.

One of the major factors that mediate the direct sequestration of PM cholesterol is sphingomyelin, which forms a complex with cholesterol and makes it inaccessible (*Endapally et al., 2019*; *Finean, 1953*; *McConnell and Radhakrishnan, 2003*; *Radhakrishnan and McConnell, 2000*; *Slotte, 1992*). The sphingomyelin-sequestered pool of PM cholesterol consists of ~15% of PM lipids, while the rest of the inaccessible pool is sequestered by other membrane factors (*Das et al., 2014*). To test whether the sequestration of cholesterol by sphingomyelin affects the binding of GRAM domains to artificial membranes, we incorporated increasing amounts of sphingomyelin (SM) (10% or 25%) into liposomes that contained 50% cholesterol and 20% phosphatidylserine (*Figure 3E*). When these liposomes contained 25% sphingomyelin, the percentage of GRAMD1b GRAM domains that bound to the liposomes decreased from ~80% to ~45% (*Figure 3E* and *Figure 3—figure supplement 1C*). Similar results were obtained with the GRAM domain of GRAMD1a (*Figure 3F*). Thus, the binding of GRAM domains to artificial

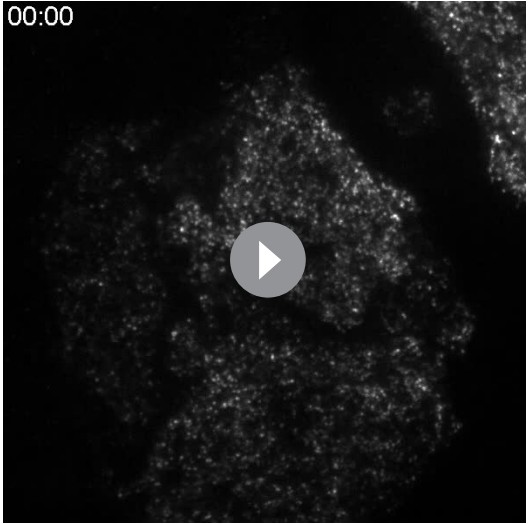

**Video 1.** GRAMD1b is recruited to ER–PM contacts upon cholesterol loading. HeLa cells expressing EGFP–GRAMD1b were imaged under TIRF microscopy. Images were taken every 20 s, and 200 µM cholesterol/MCD was added at the 5 min time point. Image size, 66.1 µm x 66.1 µm.
https://elifesciences.org/articles/51401#video1

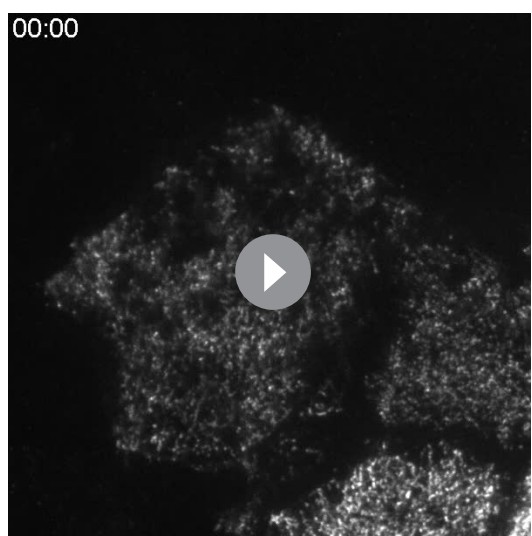

**Video 2.** GRAMD1b ΔGRAM is not recruited to ER–PM contacts upon cholesterol loading. HeLa cells expressing EGFP–GRAMD1b ΔGRAM were imaged under TIRF microscopy. Images were taken every 20 s, and 200 μM cholesterol/MCD was added at the 5 min time point. Image size, 66.1 μm x 66.1 μm.
https://elifesciences.org/articles/51401#video2

membranes that contain cholesterol and phosphatidylserine can be modulated by the presence of sphingomyelin (*Figure 3—figure supplement 1C*). These results suggest that sphingomyelin helps to suppress the binding of GRAM domains to the PM at rest by reducing the accessibility of the cholesterol in this bilayer.

In addition to sphingomyelin, phospholipid acyl chain saturation has profound effects on the accessibility of cholesterol in membranes (*Chakrabarti et al., 2017*; *Gay et al., 2015*; *Lange et al., 2013*; *Radhakrishnan and McConnell, 2000*; *Sokolov and Radhakrishnan, 2010*). If the GRAM domain binds to the PM by sensing the accessibility of cholesterol, its binding to artificial membranes should also be influenced by the acyl chain diversity of the phospholipids. To test this possibility, we generated liposomes containing fixed amounts of phosphatidylserine (20%) with varying ratios of cholesterol and phosphatidylcholine (*Figure 3—figure supplement 2B*). We individually tested the three types of phosphatidylcholine that possesses different acyl chain structures, namely POPC, DOPC, and DPhyPC (*Figure 3—figure supplement 2C*). Branched (DPhyPC) and more unsaturated (DOPC) acyl chains lower the tendency to form

ordered conformation in the membranes, and thus, POPC has the strongest cholesterol sequestration effect of these three lipids, followed by DOPC and DPhyPC (*Sokolov and Radhakrishnan, 2010*). The binding of the GRAM domain of GRAMD1b to liposomes shifted to lower cholesterol concentration as the ordering tendency of phosphatidylcholine is lowered (i.e. as the cholesterol sequestration effect is reduced) (*Figure 3—figure supplement 2B*). These results are consistent with the ability of the GRAM domain to sense the accessibility of cholesterol in membranes.

Finally, to determine whether GRAM domains bind more broadly to other anionic lipids, we replaced phosphatidylserine with other anionic lipids, namely phosphatidic acid (PA), PI(4)P, and PI(4,5)P$_2$, and asked whether they affected GRAM domain binding similarly. In this assay, we used 5% anionic lipids, including phosphatidylserine, because even 5% phosphatidylserine was sufficient to mediate the binding of the GRAM domain to liposomes that also contained 50% free cholesterol, albeit less efficiently than 20% phosphatidylserine (*Figure 3—figure supplement 1D*). No or little binding was observed when GRAM domains were mixed with liposomes that contained 5% of these anionic lipids (each was tested individually) (*Figure 3—figure supplement 1D*). However, as seen when phosphatidylserine and cholesterol were combined, the addition of free cholesterol to these anionic-lipid-containing liposomes enhanced the binding of GRAM domains to the liposomes (*Figure 3—figure supplement 1D*).

As anionic lipids, including phosphatidylserine, are enriched in the inner leaflet of the PM (*Yeung et al., 2008*), these results indicate that the recruitment of GRAMD1s to the PM is regulated by interactions between GRAM domains and anionic lipids, and that these interactions are enhanced by the additional presence of accessible/unsequestered cholesterol in the PM.

## Liberation of sphingomyelin-sequestered pool of cholesterol induces acute recruitment of GRAMD1b to the PM

To examine the physiological role of sphingomyelin in GRAM domain-dependent recruitment of GRAMD1s to the PM, HeLa cells expressing either EGFP–GRAMD1b or EGFP–GRAMD1b ΔGRAM were treated with sphingomyelinase, which hydrolyzes PM sphingomyelin, and imaged under total internal reflection fluorescence (TIRF) microscopy. Although sphingomyelin is enriched in the outer leaflet of the PM bilayer, it also contributes to suppressing the accessibility of cholesterol in the inner

leaflet of the PM, because unsequestered cholesterol can spontaneously flip flop between the outer and inner leaflets of this bilayer (*Leventis and Silvius, 2001*; *Steck and Lange, 2018*). Within 30 min of sphingomyelinase treatment, GRAMD1b was indeed recruited to the PM (*Figure 3G*), albeit this recruitment was less efficient than cholesterol loading to the PM (*Figure 3B*). GRAMD1b ΔGRAM, however, failed to localize to the PM, even after 60 min (*Figure 3G*). EGFP-tagged GRAMD1 GRAM domains (namely EGFP–GRAM$_{1a}$, EGFP–GRAM$_{1b}$, and EGFP–GRAM$_{1c}$) were all recruited to the PM upon sphingomyelinase treatment (*Figure 3H*), revealing a direct role of the GRAM domain in detecting the unsequestered/accessible pool of PM cholesterol in cells. These results are also consistent with the lack of enrichment of GRAMD1s at ER–PM contact sites at rest (*Figure 1B*, *Figure 3A* and *Figure 4—figure supplement 3B*). Taken together, these data demonstrate that GRAMD1s are recruited to the PM by sensing an increase in the accessibility of PM cholesterol (i.e. acute expansion of the accessible pool of PM cholesterol that exceeds a certain threshold at which the GRAM domain interacts with the PM). Furthermore, the data show that this recruitment depends on the GRAM domain, which acts as a coincidence detector for both unsequestered/accessible cholesterol and anionic lipids in the PM (*Figure 3I*).

## Deletion of GRAMD1s results in exaggerated accumulation of the accessible pool of cholesterol in the PM

As GRAMD1s move to ER–PM contact sites upon acute expansion of the accessible pool of PM cholesterol (*Figure 3G,H*), they may also contribute to the extraction of accessible PM cholesterol in order to maintain homeostasis. To investigate the potential functions of GRAMD1s in this process, we used the CRISPR/Cas9 system to disrupt GRAMD1 function by targeting all three GRAMD1 genes (GRAMD1A, GRAMD1B and GRAMD1C) in HeLa cells. Guide RNAs specific to exon 13 of GRAMD1A and GRAMD1B and to exon 11 of GRAMD1C were chosen, as they encode the lipid-harboring StART-like domains (*Figure 4A*). After transfection of plasmids expressing GRAMD1-specific guide RNAs and Cas9 protein, two independent isolates of GRAMD1a/1b double knockout cell clones (DKO #38 and DKO #40) and two independent isolates of GRAMD1a/1b/1c triple knockout cell clones (TKO #1 and TKO #15) were selected. The absence of GRAMD1a and GRAMD1b was confirmed by western blotting and genomic sequencing (*Figure 4B* and *Figure 4—figure supplement 1A–D*). Disruption of the GRAMD1C gene was validated by sequencing the targeted genomic region within the GRAMD1C locus (*Figure 4C* and *Figure 4—figure supplement 1E*). No obvious defects in cell viability or overall morphology were observed for these KO cells, with the exception that KO cells grew slightly slower than parental HeLa cells. Subsequent experiments were performed using GRAMD1a/1b/1c TKO #15 cells (hereafter referred to as GRAMD1 TKO cells).

The incubation of cells with sphingomyelinase reduces the sequestration of PM cholesterol, resulting in a transient expansion of the accessible pool of cholesterol in the PM (*Das et al., 2014*; *Endapally et al., 2019*). The newly expanded pool of accessible cholesterol is then extracted and transported to the ER (*Das et al., 2014*; *Lange and Steck, 1997*; *Scheek et al., 1997*; *Slotte and Bierman, 1988*). Based on the ability of the GRAM domain to sense expansion of the accessible pool of PM cholesterol (*Figure 3H,I*), EGFP–GRAM$_{1b}$ was used as a probe to detect acute increases in the accessible pool of PM cholesterol. Without stimulation, cytosolically expressed EGFP–GRAM$_{1b}$ was distributed throughout the cytoplasm without particular enrichment in the PM in both wild-type and GRAMD1 TKO HeLa cells (*Figure 4—figure supplement 2A*). Treatment with sphingomyelinase for 1 hr led to only modest recruitment of EGFP–GRAM$_{1b}$ to the PM in wild-type HeLa cells (*Figure 4D*). By contrast, the same treatment lead to much more prominent recruitment of EGFP–GRAM$_{1b}$ to the PM in GRAMD1 TKO cells (*Figure 4D*). TIRF microscopy of cells expressing EGFP–GRAM$_{1b}$ revealed that PM recruitment of EGFP–GRAM$_{1b}$ upon sphingomyelinase treatment was significantly enhanced in GRAMD1 TKO cells, compared to wild-type control cells, over the entire 1 hr treatment (*Figure 4E*). Importantly, additional treatment of GRAMD1 TKO cells with methyl-β-cyclodextrin (MCD), which extracts cholesterol from cellular membranes, resulted in acute loss of the PM recruitment of EGFP–GRAM$_{1b}$ within 2 min (*Figure 4—figure supplement 2B*). However, the same treatment resulted in only modest changes in the binding of the phosphatidylserine biosensor (the mCherry-tagged C2 domain of lactadherin, mCherry-LactC2) or the PI(4,5)P$_2$ biosensor (the iRFP-tagged PH domain of PLCδ, iRFP-PH$^{PLCδ}$), confirming the specificity of EGFP–GRAM$_{1b}$ in sensing the newly expanded pool of accessible cholesterol in the PM upon sphingomyelinase treatment (*Figure 4—figure supplement 2B*). Taken together, these results demonstrate an

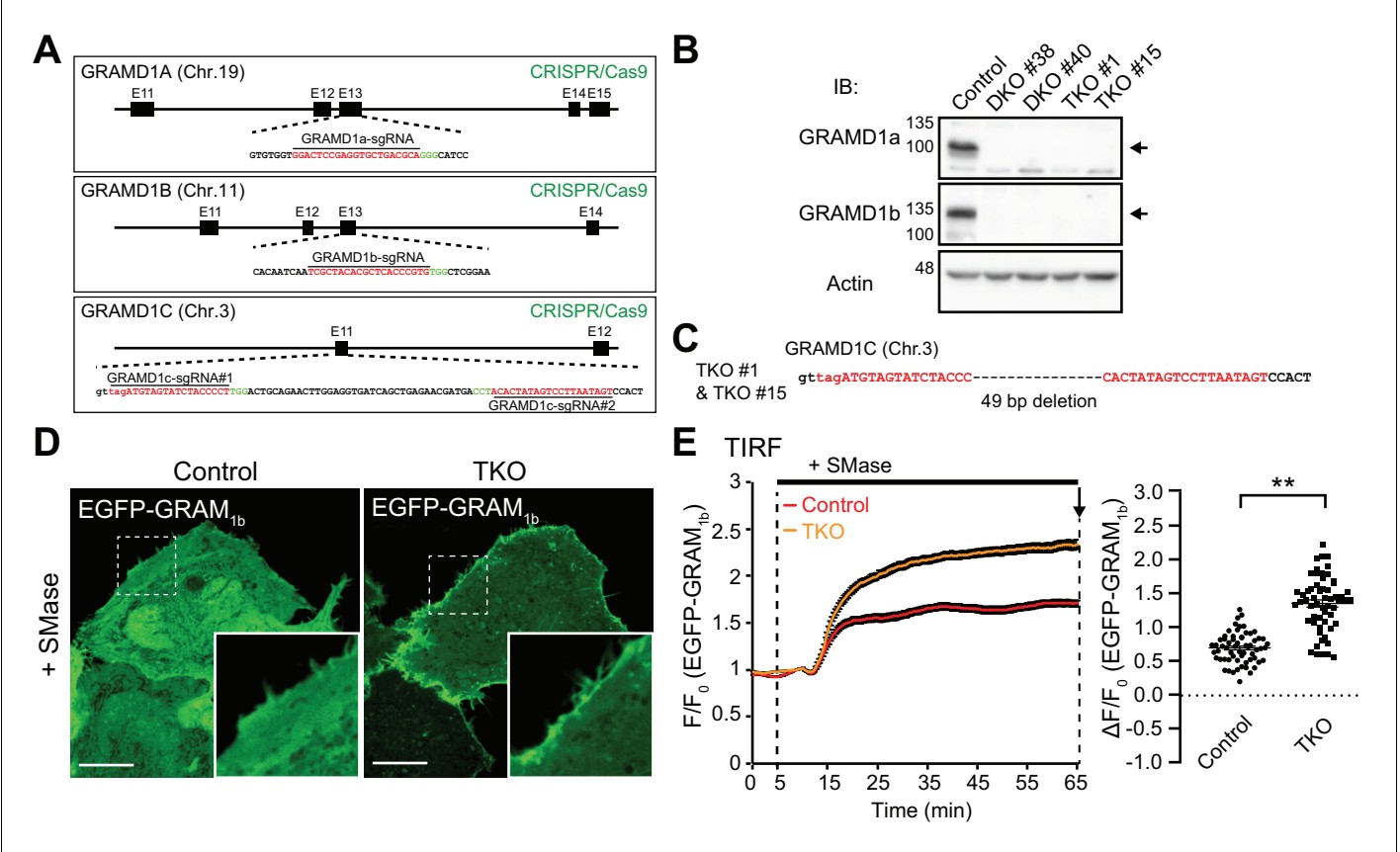

**Figure 4.** Deletion of GRAMD1s results in exaggerated accumulation of the accessible pool of cholesterol in the PM. (A) Schematics of the Cas9/sgRNA targeting sites in human GRAMD1A, GRAMD1B and GRAMD1C loci. The targeting sequences are highlighted in red. The protospacer-adjacent motifs (PAMs) are labeled in green. (B) Lysates of control HeLa cells, two independently isolated GRAMD1a/1b DKO cell lines, and two independently isolated GRAMD1 TKO cell lines were processed by SDS–PAGE and immunoblotting (IB) with anti-GRAMD1a, anti-GRAMD1b and anti-actin antibodies. The arrows indicate the specific bands for GRAMD1a and GRAMD1b. (C) Nucleotide sequence analysis of the GRAMD1C gene of the GRAMD1 TKO cell lines. Guide RNA-targeting sites are highlighted in red. (D) Confocal images of live wild-type (Control) and GRAMD1 TKO (TKO) HeLa cells expressing the EGFP-tagged GRAM domain of GRAMD1b (EGFP–GRAM$_{1b}$) with SMase treatment (100mU/ml for 1 hr at 37°C). Insets show at higher magnification the regions indicated by white dashed boxes. Note the strong recruitment of EGFP–GRAM$_{1b}$ to the PM of GRAMD1 TKO cells compared to that of the control cells. Scale bars, 10 μm. (E) Left: time course of normalized EGFP signal, as assessed by TIRF microscopy, from wild-type (Control) and GRAMD1 TKO (TKO) HeLa cells expressing EGFP–GRAM$_{1b}$. SMase treatment (100 mU/ml) is indicated. Right: values of ΔF/F$_0$ corresponding to the end of the experiment as indicated by the arrow [mean ± SEM, n = 62 cells (Control), n = 58 cells (TKO); data are pooled from three independent experiments for each condition; two-tailed unpaired Student's t-test with equal variance, **p<0.0001].

The online version of this article includes the following source data and figure supplement(s) for figure 4:

**Source data 1.** Dataset for *Figure 4*.

**Figure supplement 1.** Generation of GRAMD1 triple knockout (TKO) HeLa cells.

**Figure supplement 2.** Recruitment of the EGFP-tagged GRAM domain of GRAMD1b (EGFP–GRAM$_{1b}$) to the PM requires cholesterol.

**Figure supplement 2—source data 1.** Dataset for *Figure 4—figure supplement 2*.

**Figure supplement 3.** Isolation and characterization of the PM sheets of GRAMD1 triple knockout (TKO) HeLa cells.

**Figure supplement 3—source data 1.** Dataset for *Figure 4—figure supplement 3*.

exaggerated accumulation of the accessible pool of PM cholesterol in GRAMD1 TKO cells upon sphingomyelinase treatment, and suggest that the extraction and transport of this acutely expanded accessible pool may be impaired in the absence of GRAMD1s.

We also assessed the role of GRAMD1s in regulating steady-state PM cholesterol levels by separating and purifying PMs from cultured cells using poly-D-lysine-coated dextran beads (*Saheki et al., 2016*). Cultured cells were attached to the beads and osmotically lysed by vigorous vortexing. Brief sonication was used to remove most organelles, whereas PM sheets remained attached to the bead surface (visualized by BODIPY-labeled ceramide) (*Figure 4—figure supplement 3A*). As shown by

western blotting, the PM sheets that remained bound to the beads were highly enriched for PM marker proteins (such as CD44) relative to the starting material. The endosomal marker, EEA1, was greatly depleted, whereas small amounts of ER proteins (such as VAPA and VAPB) were recovered in the PM (*Figure 4—figure supplement 3B*). This probably reflected the tight attachment of cortical ER (*Saheki et al., 2016*). Importantly, the levels of endogenous GRAMD1a and GRAMD1b on bead-attached PM sheets were similar to those seen for the integral ER protein, VAP (*Figure 4—figure supplement 3B*). This confirmed that the majority of these two proteins are distributed throughout the ER, with only a very small fraction localizing to ER–PM contact sites at rest (*Figure 1B* and *Figure 3A*). Mass spectrometry analysis of whole-cell and purified PM lipid extracts from wild-type control and GRAMD1 TKO HeLa cells did not reveal significant changes in cholesterol and other major lipids, except for very minor increases in cholesterol esters (*Figure 4—figure supplement 3C, D*). Thus, GRAMD1s are not essential for maintaining total levels of PM cholesterol. This result is also consistent with very little enrichment of GRAMD1s at ER–PM contacts at the steady state. Collectively, these results indicate that GRAMD1s may contribute to PM cholesterol homeostasis by counteracting acute increases in the accessible pool of PM cholesterol through its extraction and transport to the ER.

## The cholesterol transporting property of the StART-like domain of GRAMD1s is critical for the removal of an acutely expanded pool of accessible PM cholesterol

Although GRAMD1 StART-like domains transport cholesterol in vitro, it remains unclear whether this property is relevant to cellular physiology. Our live-cell imaging analysis of EGFP–GRAM$_{1b}$ (which is a novel biosensor for detecting acute expansion of the accessible pool of PM cholesterol that we identified in this study) allowed us to conduct a structure–function analysis of GRAMD1s in the context of cellular functions for the first time. We first asked whether the sterol-binding pocket of the StART-like domain is required for the cellular functions of GRAMD1s. As a first step, we characterized the cholesterol-transporting properties of individual StART-like domains in vitro and generated a series of structure-guided mutations in order to identify key amino-acid residues that are essential for cholesterol transport. We purified StART-like domains from all three GRAMD1s and performed cell-free liposome-based lipid transfer assays. In this assay, the amount of dehydroergosterol (DHE) (a fluorescent analog of cholesterol) in liposomes was quantitatively measured using fluorescence resonance energy transfer (FRET) between DHE and Dansyl-PE (DNS-PE) (*Figure 5A*). DHE was initially loaded only into donor liposomes, and its transfer from donor to DNS-PE-containing acceptor liposomes was monitored over time by measuring FRET between transferred DHE and DNS-PE in acceptor liposomes (*Figure 5A* and *Figure 5—figure supplement 1A*). In the absence of StART-like domains, very few increases in the FRET signal were observed (*Figure 5—figure supplement 1B*, buffer). However, when GRAMD1 StART-like domains were mixed with donor and acceptor liposomes, a rapid increase in FRET signal was observed, indicating the efficient extraction of DHE from donor liposomes and its loading onto acceptor liposomes by the StART-like domains (*Figure 5E* and *Figure 5—figure supplement 1B*). Increasing amounts of purified proteins (0.5 μM, 1 μM, and 2 μM) reduced the time required for the FRET signal to plateau (*Figure 5—figure supplement 1C–E*). GRAMD1a StART-like domains transferred DHE most efficiently, at a rate corresponding to ~8 DHE molecules per minute. In comparison, GRAMD1b and GRAMD1c transported ~1 DHE molecule per minute, as calculated using a standard curve (*Figure 5—figure supplement 1A,F*). Our results show the ability of GRAMD1 StART-like domains to transport cholesterol between membranes.

Guided by the crystal structures of GRAMD1 StART-like domains in complex with 25-hydroxycholesterol (*Laraia et al., 2019*; *Sandhu et al., 2018*), we designed mutations that would potentially block the insertion of cholesterol into the GRAMD1b StART-like domain. Our mutagenesis strategy was to rigidify the loop that was predicted to open or close to capture or release sterol (5P) (*Figure 5B*). Purified GRAMD1a and GRAMD1b StART-like domains with 5P mutations were unable to transfer DHE in vitro (*Figure 5C,D* and *Figure 5—figure supplement 1G,H*). A similar result was also obtained with a version of the GRAMD1b StART-like domain with a point mutation (T469D) that was previously shown to be defective in DHE extraction in vitro (*Horenkamp et al., 2018*) (*Figure 5—figure supplement 1H*).

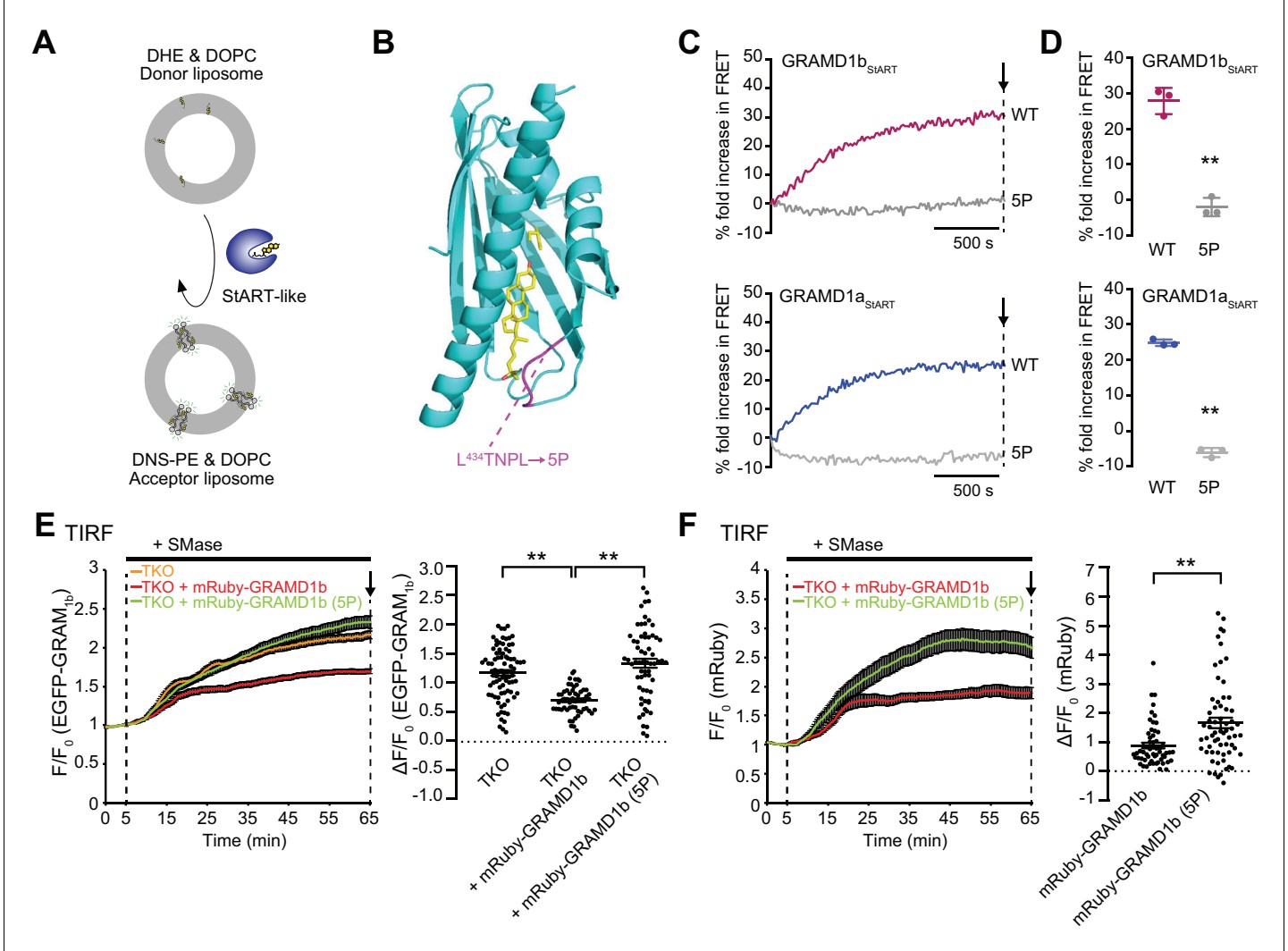

**Figure 5.** The cholesterol transporting property of the StART-like domain of GRAMD1s is critical for removal of an acutely expanded pool of accessible PM cholesterol. (**A**) Schematic showing the design of the in vitro lipid transfer assay. Donor liposomes (10% DHE, 90% DOPC) and acceptor liposomes [2.5% Dansyl-PE (DNS-PE), 97.5% DOPC] were incubated with the purified StART-like domain of GRAMD1s (GRAMD1a$_{StART}$, GRAMD1b$_{StART}$, or GRAMD1c$_{StART}$). Transfer of DHE from donor to acceptor liposomes, which results in an increase in fluorescence resonance energy transfer (FRET) between DHE and DNS-PE in acceptor liposomes, was monitored using a fluorometer (see Materials and methods). (**B**) Design of a mutant StART-like domain that is defective in lipid harboring. Ribbon diagram of the modeled GRAMD1b$_{StART}$ (see Materials and methods) with designed mutations (5P) in the Ω1 loop of GRAMD1b, which is predicted to open and close to capture sterol. (**C, D**) 5P mutations in the Ω1 loop of the StART-like domain impairs DHE transfer activity. (**C**) Time course of fold increase in FRET signals. WT GRAMD1b$_{StART}$ and GRAMD1b$_{StART}$ with 5P mutation (2 μM, top panel) and WT GRAMD1a$_{StART}$ and GRAMD1a$_{StART}$ with 5P mutation (0.5 μM, bottom panel) were individually added at time 0. (**D**) Values of fold increase in FRET signals of acceptor liposomes in the presence of the indicated proteins at the time point corresponding to the end of the experiments [as shown by arrows in (C)] (mean ± SEM, n = 3 independent experiments for all of the conditions; two-tailed unpaired Student's t-test, GRAMD1b$_{StART}$**p=0.0003, GRAMD1a$_{StART}$**p<0.0001). (**E, F**) Left: time course of normalized (**E**) EGFP or (**F**) mRuby signal, as assessed by TIRF microscopy, from GRAMD1 TKO cells expressing EGFP–GRAM$_{1b}$ and mRuby-tagged constructs as indicated. SMase treatment (100 mU/ml) is indicated. Right: values of ΔF/F$_0$ at the time point corresponding to the end of the experiment (as indicated by the arrows). [(E) mean ± SEM, n = 84 cells (TKO), n = 57 cells (TKO + mRuby–GRAMD1b), n = 64 cells (TKO + mRuby–GRAMD1b (5P)); Tukey's multiple comparisons test, **p<0.0001; (F) mean ± SEM, n = 57 cells (TKO + mRuby–GRAMD1b), n = 64 cells (TKO + mRuby–GRAMD1b (5P)); two-tailed unpaired Student's t-test, **p=0.0003; data are pooled from three or four independent experiments for each condition.]

The online version of this article includes the following source data and figure supplement(s) for figure 5:

**Source data 1.** Dataset for *Figure 5*.
**Figure supplement 1.** Characterization of the cholesterol transporting property of the StART-like domain of GRAMD1s.
**Figure supplement 1—source data 1.** Dataset for *Figure 5—figure supplement 1*.
*Figure 5 continued on next page*

*Figure 5 continued*

**Figure supplement 2.** Overexpression of STARD4 and selected ORPs does not rescue the exaggerated accumulation of the accessible pool of PM cholesterol in GRAMD1 TKO cells upon sphingomyelinase treatment.

**Figure supplement 2—source data 1.** Dataset for *Figure 5—figure supplement 2*.

Building upon our newly designed 5P mutation, which eliminated the ability of StART-like domains to transport cholesterol, we asked whether the exaggerated accumulation of the accessible pool of PM cholesterol that was observed in GRAMD1 TKO cells upon sphingomyelinase treatment (using the EGFP–GRAM$_{1b}$ biosensor) could be rescued by re-expressing wild-type or mutant versions of GRAMD1b. Strikingly, the enhanced PM recruitment of EGFP–GRAM$_{1b}$ was dramatically suppressed by expressing wild-type mRuby–GRAMD1b but not by expressing a mutant version of mRuby–GRAMD1b that is defective in cholesterol transport [mRuby–GRAMD1b (5P)] (*Figure 5E*). By contrast, PM recruitment of mRuby–GRAMD1b upon sphingomyelinase treatment of TKO cells was higher for the 5P mutant GRAMD1b than for wild-type GRAMD1b (*Figure 5F*). These results suggest that the StART-like domain-dependent extraction and transport of accessible PM cholesterol to the ER facilitates the dissociation of GRAMD1b from the PM, as the interaction of the GRAM domain of GRAMD1b with the PM is weakened, owing to a reduction in accessible cholesterol in the PM.

Our results to date suggest that GRAMD1b may play a unique role in sensing and controlling the movement of accessible PM cholesterol. To further support this notion, we used GRAMD1 TKO cells to examine whether overexpression of other known cholesterol-transfer proteins, such as STARD4 (*Iaea et al., 2017*; *Mesmin et al., 2011*) and some ORPs [including OSBP (*Antonny et al., 2018*), ORP4 (*Charman et al., 2014*) and ORP9 (*Ngo and Ridgway, 2009*)] could substitute the function of GRAMD1s. Specifically, we examined whether their overexpression rescue exaggerated accumulation of the accessible pool of PM cholesterol observed in GRAMD1 TKO cells, as monitored by the EGFP–GRAM$_{1b}$ biosensor, upon sphingomyelinase treatment (*Figure 4D,E*). Transiently transfected mCherry-tagged STARD4 (mCherry–STARD4) and mRuby-tagged ORPs (mRuby–OSBP, mRuby–ORP4, and mRuby–ORP9) were all well expressed in TKO cells (*Figure 5—figure supplement 2A*). However, their expression did not suppress the enhanced recruitment of EGFP–GRAM$_{1b}$ to the PM in TKO cells upon sphingomyelinase treatment, being unable to substitute the function of GRAMD1s (*Figure 5—figure supplement 2B,D*; compare with *Figure 5E*). None of these proteins were recruited to the PM by sphingomyelinase treatment, demonstrating a unique property of GRAMD1s in sensing a transient expansion of the accessible pool of PM cholesterol (*Figure 5—figure supplement 2C,E*).

Taken together, our results suggest a critical role of the GRAMD1s in controlling the movement of the accessible pool of PM cholesterol between the PM and the ER via their StART-like domains.

## GRAMD1s play a role in accessible cholesterol transport from the PM to the ER during acute expansion of the accessible pool of PM cholesterol

Acute expansion of the accessible pool of PM cholesterol results in the suppression of SREBP-2 cleavage and the inhibition of cholesterol biosynthesis as a result of transport of accessible cholesterol from the PM to the ER. However, the intracellular transport machinery by which accessible cholesterol is transported from the PM to the ER remains unknown. GRAMD1s may play a role in this process, as they are able to sense and counteract the acute expansion of the accessible pool of PM cholesterol.

TIRF microscopy of cells expressing EGFP–GRAMD1b revealed that sphingomyelinase treatment led to sustained recruitment of GRAMD1b to the PM (during 3 hr of imaging) (*Figure 6A*; *Video 3*). As GRAMD1 TKO cells show exaggerated accumulation of the accessible pool of PM cholesterol upon sphingomyelinase treatment compared with wild-type cells (*Figure 4D,E*), GRAMD1s may be involved in PM to ER transport of the accessible pool of cholesterol via their GRAM and StART-like domains. To examine the role of GRAMD1s in this process, we determined a time-course for the suppression of SREBP-2 cleavage upon sphingomyelinase treatment in wild-type control and GRAMD1 TKO cells as an estimate of the efficiency of the transport of accessible cholesterol from the PM to the ER. In this assay, we first depleted most of the accessible cholesterol from control and

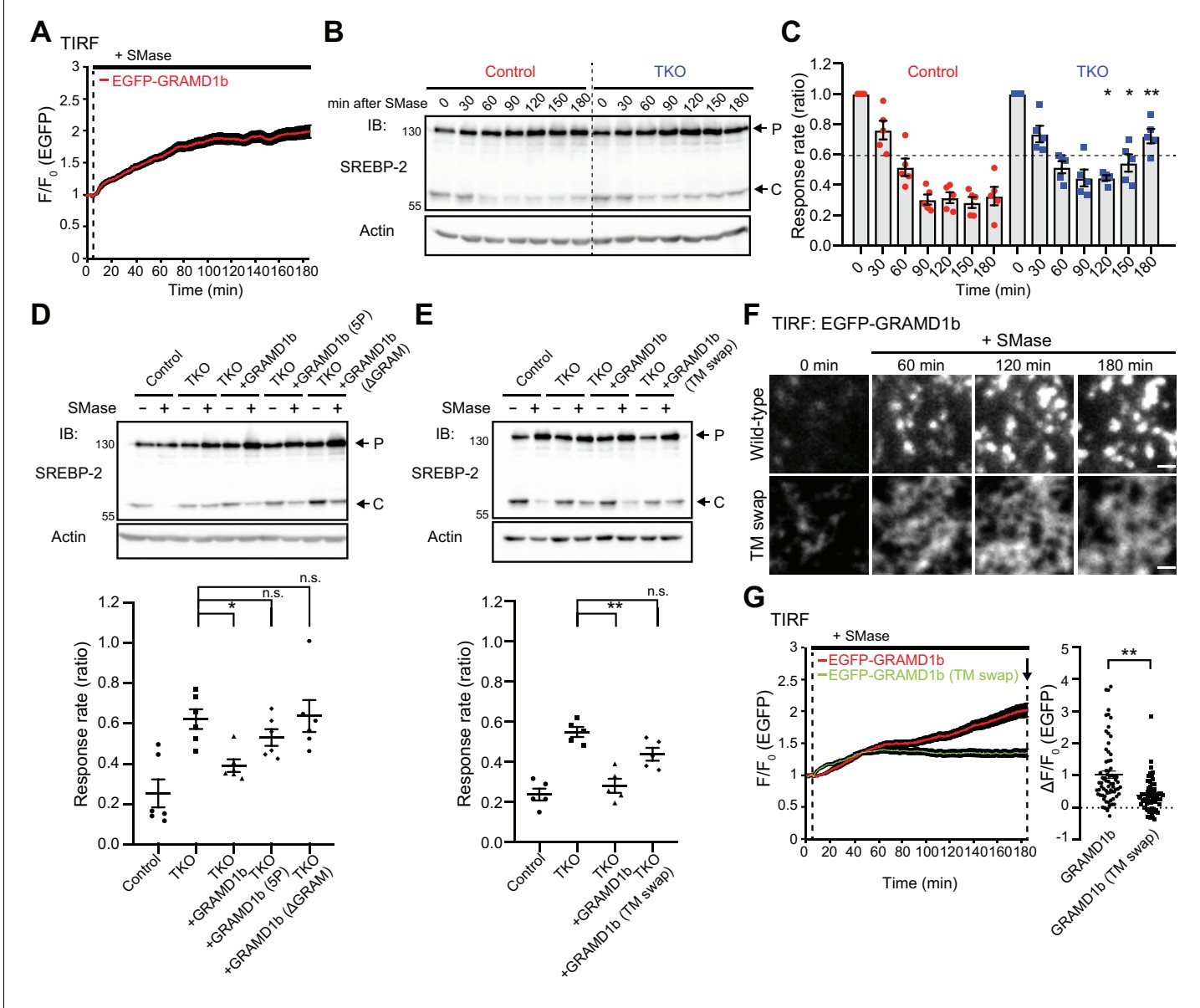

**Figure 6.** GRAMD1s-mediated PM to ER cholesterol transport plays a role in the suppression of SREBP-2 cleavage upon sphingomyelinase treatment. (**A**) Time course of normalized EGFP signal, as assessed by TIRF microscopy, from HeLa cells expressing EGFP–GRAMD1b in response to sphingomyelinase (SMase) treatment (100 mU/ml). Note the sustained recruitment of EGFP–GRAMD1b to the PM even after 3 hr of the SMase treatment (mean ± SEM, n = 74 cells; data are pooled from four independent experiments). (**B**) Wild-type (Control) and GRAMD1 TKO (TKO) HeLa cells were cultured in the medium supplemented with 10% lipoprotein-deficient serum (LPDS) and mevastatin (50 μM) for 16 hr and then treated with SMase (100 mU/ml) for the indicated time at 37˚C. Lysates of the cells were processed for SDS-PAGE and immunoblotting (IB) with anti-SREBP-2 and anti-Actin antibodies. Arrows indicate precursor (P) and cleaved (C) forms of SREBP-2, respectively. (**C**) Quantification of the response rate of the suppression of SREBP-2 cleavage upon SMase treatment from the experiment shown in (**B**). For each time point, the ratio of the band intensity of the cleaved SREBP-2 over the total band intensity of cleaved and precursor forms of SREBP-2 was normalized by the ratio obtained from time 0, and plotted as response rate. Note that the suppression of SREBP-2 cleavage is attenuated in GRAMD1 TKO cells [mean ± SEM, n = 5 lysates (independent experiments) for each time point; multiple comparisons were made using the Holm-Sidak method, *p=0.0461 (120 min), *p=0.0238 (150 min), **p=0.0052 (180 min)]. (**D**, **E**) Wild-type (Control) and GRAMD1 TKO (TKO) HeLa cells, transfected with the EGFP-tagged GRAMD1s constructs as indicated, were cultured in the medium supplemented with 10% lipoprotein-deficient serum (LPDS) and mevastatin (50 μM) for 16 hr and then treated with SMase (100 mU/ml) for 3 hr at 37˚C. Top: lysates of the cells were processed for SDS-PAGE and IB with anti-SREBP-2 and anti-actin antibodies. Arrows indicate precursor (P) and cleaved (C) forms of SREBP-2. Bottom: the response rate was calculated as in panel (**C**) except that the ratio obtained from the cells with SMase treatment was normalized by the ratio obtained from the cells without SMase treatment for each condition. Note the rescue by expression of EGFP–GRAMD1b but not by mutant versions of EGFP–GRAMD1b (5P, ΔGRAM, TM swap) [(**D**) mean ± SEM, n = 6 lysates (independent experiments) for each

*Figure 6 continued on next page*

*Figure 6 continued*

condition, Dunnett's multiple comparisons test, *p=0.0162; (E): mean ± SEM, n = 5 lysates (independent experiments) for each condition, Dunnett's multiple comparisons test, **p<0.0001; n.s. denotes not significant]. (F) Representative TIRF images of live HeLa cells expressing EGFP–GRAMD1b (Wild-type) and EGFP–GRAMD1b TM swap (TM swap) treated as described for panel (G). Note the differences in how these proteins are recruited to the PM. Wild-type GRAMD1b accumulated progressively at ER–PM contacts, forming patches by the end of the 3 hr imaging period, whereas the GRAMD1b TM swap mutant remained diffuse on the tubular ER, even at the end of the 3 hr imaging period. Scale bars, 1 μm. (G) Left: time course of normalized EGFP signal, as assessed by TIRF microscopy, from HeLa cells expressing EGFP–GRAMD1b or EGFP–GRAMD1b TM swap in response to SMase treatment (100 mU/ml). Note the reduced recruitment of EGFP–GRAMD1b TM swap to the PM compared to EGFP–GRAMD1b after 3 hr of the SMase treatment. Right: values of $\Delta F/F_0$ at a time point corresponding to the end of the experiment (as indicated by the arrow) [mean ± SEM, n = 72 cells (GRAMD1b), n = 69 cells (GRAMD1b TM swap); data are pooled from three independent experiments for each condition; two-tailed unpaired Student's t-test, **p<0.0001].

The online version of this article includes the following source data and figure supplement(s) for figure 6:

**Source data 1.** Dataset for *Figure 6*.
**Figure supplement 1.** Transport of cholesterol from the PM to the ER by GRAMD1 proteins requires their StART-like and GRAM domains as well as their complex formation.
**Figure supplement 1—source data 1.** Dataset for *Figure 6—figure supplement 1*.
**Figure supplement 2.** Deletion of GRAMD1s results in sustained D4 binding to the PM upon sphingomyelinase treatment.
**Figure supplement 2—source data 1.** Dataset for *Figure 6—figure supplement 2*.

TKO cells by treating them with a combination of lipoprotein-deficient serum (LPDS) and mevastatin, an HMG-CoA reductase inhibitor, for 16 hr (a treatment designed to induce maximum SREBP-2 cleavage by cholesterol starvation). We then stimulated the cells with sphingomyelinase and, using total cell lysates, we monitored over time the suppression of SREBP-2 cleavage, which results from the PM to ER transport of accessible cholesterol in response to the liberation of the sphingomyelin-sequestered pool of PM cholesterol by sphingomyelinase. Cell lysates were collected at different time points (0, 30, 60, 90, 120, 150, and 180 min) and analyzed by SDS-PAGE followed by immuno-blotting against SREBP-2 (*Figure 6B*). At time 0, there were no detectable changes in the cleavage of SREBP-2 in GRAMD1 TKO cells compared to wild-type control cells. Suppression of SREBP-2 cleavage was observed in control cell lysates within 90 min; however, such suppression was delayed and reduced (but not eliminated) in TKO cells. Even after 180 min, TKO cells were not able to suppress SREBP-2 cleavage to levels similar to those observed in wild-type control cells (*Figure 6B,C*).

Importantly, re-expression of GRAMD1b in TKO cells was sufficient to suppress SREBP-2 cleavage to an extent similar to that observed in wild-type control cells at the 180 min time point, thereby rescuing the phenotype (*Figure 6D,E* and *Figure 6—figure supplement 1A,B*). We hypothesized that both the recruitment of GRAMD1s to ER–PM contact sites and their ability to transport cholesterol are critical for the suppression of the cleavage of SREBP-2 by facilitating transport of the newly expanded pool of accessible PM cholesterol to the ER. To test this hypothesis, we used a GRAMD1b mutant that lacks the GRAM domain (GRAMD1b ΔGRAM), which cannot be recruited to the PM (*Figure 3G*), and a GRAMD1b with the mutated StART-like domain, which is defective in cholesterol transport (5P) (*Figure 5B–D*). The expression of GRAMD1b ΔGRAM or GRAMD1b 5P in TKO cells failed to rescue the phenotype (*Figure 6D* and *Figure 6—figure supplement 1A*). These data demonstrate that GRAMD1s play a role in the transport of accessible cholesterol from the PM to the ER upon acute expansion of the accessible pool of PM cholesterol and help to suppress SREBP-2 activity. Furthermore, the data show that such functions require the recruitment of GRAMD1s to ER–PM contact sites, which is

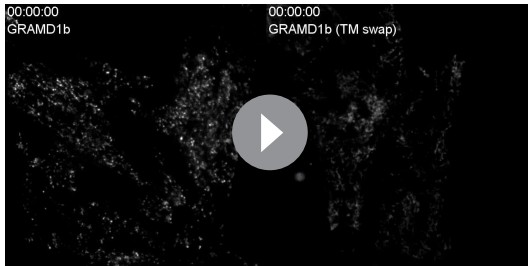

**Video 3.** Comparison of the recruitment to the PM of a wild-type GRAMD1b and of a mutant version of GRAMD1b that is defective in complex formation upon sphingomyelinase treatment. HeLa cells expressing (left) EGFP–GRAMD1b or (right) EGFP–GRAMD1b TM swap were imaged under TIRF microscopy. Images were taken every 20 s, and 100 mU/ml of sphingomyelinase (SMase) was added at the 10 min time point. Image size, 66.1 μm x 66.1 μm.
https://elifesciences.org/articles/51401#video3

regulated by the ability of their GRAM domain to sense a transient expansion of the accessible pool of PM cholesterol, and by their StART-like-domain-dependent cholesterol transport.

In order to measure changes in the accessible pool of PM cholesterol, we took advantage of the cholesterol-binding domain 4 (D4) of bacterial Perfringolysin O (PFO), which has been widely used as a probe to measure the accessible pool of PM cholesterol (*Das et al., 2013*; *Gay et al., 2015*; *Shimada et al., 2002*; *Sokolov and Radhakrishnan, 2010*). Wild-type control and GRAMD1 TKO cells that had been pre-treated with a combination of LPDS and mevastatin for 16 hr were stimulated with sphingomyelinase for a fixed period of time (0, 30, 60, 90, 120, 150, and 180 min) and washed. Cells were then incubated with recombinant EGFP-tagged D4 (EGFP–D4) proteins for 15 min at room temperature. After washing, cell lysates were collected and analyzed by SDS-PAGE followed by immuno-blotting against GFP to detect EGFP–D4 proteins that were bound to accessible cholesterol in the PM (*Figure 6—figure supplement 2A*). At time 0, there were no detectable changes in EGFP–D4 signals in TKO cells compared to wild-type control cells. A 30 min treatment with sphingomyelinase induced a similar increase in the binding of EGFP–D4 to both control and TKO cells. A gradual decrease of EGFP–D4 signals was observed in control cell lysates over the time course of 180 min, similar to that reported in a previous report that utilized a mutant form of PFO to assess changes in accessible cholesterol in the PM upon sphingomyelinase treatment (*Das et al., 2014*). TKO cells, however, showed continuous increase in binding of EGFP–D4 to the PM even after 180 min (*Figure 6—figure supplement 2A,B*), suggesting a sustained accumulation of accessible cholesterol in the PM of TKO cells, due to less efficient transport of accessible cholesterol from the PM to the ER, that does not occur in wild-type control cells.

Together with the results obtained with the cytosolically expressed EGFP–GRAM$_{1b}$ biosensor (*Figure 4D,E*), these data strongly indicate that the extraction and transport of accessible PM cholesterol to the ER by GRAMD1s is able to counteract with acute expansion of the accessible pool of PM cholesterol (e.g. acute expansion induced by sphingomyelinase treatment) to prevent the accumulation of accessible cholesterol in the PM in wild-type control cells, and that this homeostatic response is impaired in GRAMD1 TKO cells. It is also important to note that there might be other intracellular cholesterol transport mechanisms that may act in parallel with GRAMD1s to facilitate accessible cholesterol extraction from the PM for its transport to the ER, as suppression of SREBP-2 cleavage is not eliminated even in the total absence of GRAMD1s (see Discussion).

## Efficient transport of the accessible pool of PM cholesterol to the ER requires GRAMD1 complex formation

A version of GRAMD1b in which the transmembrane domain and luminal region are both replaced by those of Sec61β (TM swap) cannot form protein complexes (*Figure 2F–J*). Remarkably, GRAMD1b TM swap failed to rescue the reduced suppression of SREBP-2 cleavage observed in GRAMD1 TKO cells (*Figure 6E* and *Figure 6—figure supplement 1B*) and failed to suppress the enhanced recruitment of EGFP–GRAM$_{1b}$ to the PM in TKO cells upon sphingomyelinase treatment, although the mutant protein was still recruited to the PM (*Figure 6—figure supplement 1C,D*). TIRF microscopy analysis of HeLa cells expressing the GRAMD1b TM swap mutant, however, revealed major differences in how this protein was recruited to the PM compared to wild-type GRAMD1b (*Figure 6F*). GRAMD1b TM swap remained diffusely distributed on the tubular ER (which is closely attached to the PM) even at the end of the 180 min imaging period. By contrast, wild-type GRAMD1b progressively accumulated at ER–PM contacts as discrete patches with much stronger PM recruitment (*Figure 6F,G*; *Video 3*). These results support an important role for GRAMD1 complex formation in facilitating the progressive accumulation of GRAMD1s at ER–PM contacts, thereby supporting efficient accessible cholesterol transport at these contacts. Taken together, we conclude that GRAMD1s play a role in PM to ER transport of the accessible pool of PM cholesterol upon acute expansion of this pool. Loss of GRAMD1 function leads to sustained accumulation of accessible cholesterol in the PM, resulting in less effective suppression of SREBP-2 cleavage and possibly dysregulation of cellular cholesterol homeostasis.

## Chronic expansion of the accessible pool of PM cholesterol in GRAMD1 TKO cells

Distinct pools of cholesterol co-exist in the PM at steady state: a major pool is 'inaccessible' (i.e., sequestered or chemically inactive) and a smaller pool is 'accessible' (i.e., unsequestered or chemically active). Given the role of GRAMD1s in facilitating the transport of accessible cholesterol from the PM to the ER, the impact of GRAMD1 deficiency on steady-state levels of accessible PM cholesterol was examined.

We purified EGFP-tagged D4 mutant (D434S) proteins (EGFP–D4H), which have a lower threshold for binding to accessible cholesterol compared to D4 in vitro (*Johnson et al., 2012*; *Maekawa and Fairn, 2015*). Wild-type control and GRAMD1 TKO HeLa cells that express a PM marker (iRFP-PH$^{PLC\delta}$) were incubated with buffer containing purified recombinant EGFP–D4H proteins for 15 min at room temperature and washed, and then imaged under spinning disc confocal microscopy. D4H binding was assessed by line scan analysis. Strikingly, EGFP–D4H proteins bound more strongly to the PM of GRAMD1 TKO cells compared to that of control cells (*Figure 7A,B*). Pre-treatment of GRAMD1 TKO cells with MCD for 30 min resulted in loss of the binding of EGFP–D4H to the PM (*Figure 7—figure supplement 1A,B*), validating the specificity of this probe in sensing the accessible pool of PM cholesterol. As the total level of PM cholesterol was not elevated in GRAMD1 TKO cells in our lipidomics analysis (*Figure 4—figure supplement 3C,D*), these results indicate that the chronic expansion of the accessible pool of PM cholesterol occurs in the absence of GRAMD1s.

Re-expression of any of the three GRAMD1s in TKO cells was sufficient to reduce the binding of EGFP–D4H to the PM, thereby rescuing the chronic expansion of the D4H-accessible pool of PM cholesterol observed in TKO cells (*Figure 7—figure supplement 2A–C*). Versions of GRAMD1b in which the StART-like domain was mutated were systematically expressed in TKO cells to determine whether the ability of GRAMD1b to transport accessible cholesterol is required to rescue the phenotype (*Figure 7—figure supplement 2D*). All mutant versions of GRAMD1b, including a newly designed mutant in which the hydrophobicity of the surface of the sterol-binding pocket is changed (Y430A, V445A), as well as 5P and T469D mutants, failed to reduce the binding of EGFP–D4H to the PM of TKO cells because they are unable to rescue the chronic expansion of the D4H-accessible pool of PM cholesterol in TKO cells (*Figure 7—figure supplement 2E–I*).

Taken together, these results suggest the importance of GRAMD1s in maintaining steady-state levels of accessible PM cholesterol by facilitating its transport from the PM to the ER.

## Acute recruitment of GRAMD1b to ER–PM contacts facilitates removal of the expanded pool of accessible PM cholesterol in GRAMD1 TKO cells

Chronic expansion of the accessible pool of PM cholesterol in GRAMD1 TKO cells at steady state, revealed by increased PM binding of the EGFP–D4H probe, indicates that GRAMD1s are important for maintaining PM cholesterol homeostasis through their functions in sensing a transient expansion of the accessible pool of PM cholesterol and by facilitating the transport of accessible PM cholesterol to the ER at ER–PM contact sites. If this is the case, artificial forced recruitment of re-expressed GRAMD1s to ER–PM contacts in GRAMD1 TKO cells should mediate the extraction and transport of accessible cholesterol from the PM to the ER and reduce the binding of the EGFP–D4H probe to the PM.

To test whether GRAMD1s can directly act at ER–PM contact sites, rapamycin-induced dimerization of the FK506-binding protein (FKBP) and the FKBP-rapamycin-binding domain (FRB) (*Muthuswamy et al., 1999*) was used to recruit GRAMD1b to these sites acutely. In this assay, GRAMD1 TKO cells were co-transfected with a version of GRAMD1b in which the N-terminus, which contains the GRAM domain, was replaced by a miRFP-tagged FKBP module (miRFP-FKBP–GRAMD1b) and a PM-targeted FRB module (PM-FRB–mCherry) (*Figure 7C*). TIRF microscopy revealed rapid recruitment of miRFP-FKBP–GRAMD1b to the PM within 10 min of rapamycin treatment (*Figure 7D* and *Figure 7—figure supplement 3A*; *Video 4*). To assess accessible pool of PM cholesterol after the acute recruitment of the chimeric GRAMD1b protein to the PM, cells that had been pre-treated with rapamycin for a fixed period of time (0 min, 30 min, and 60 min) were

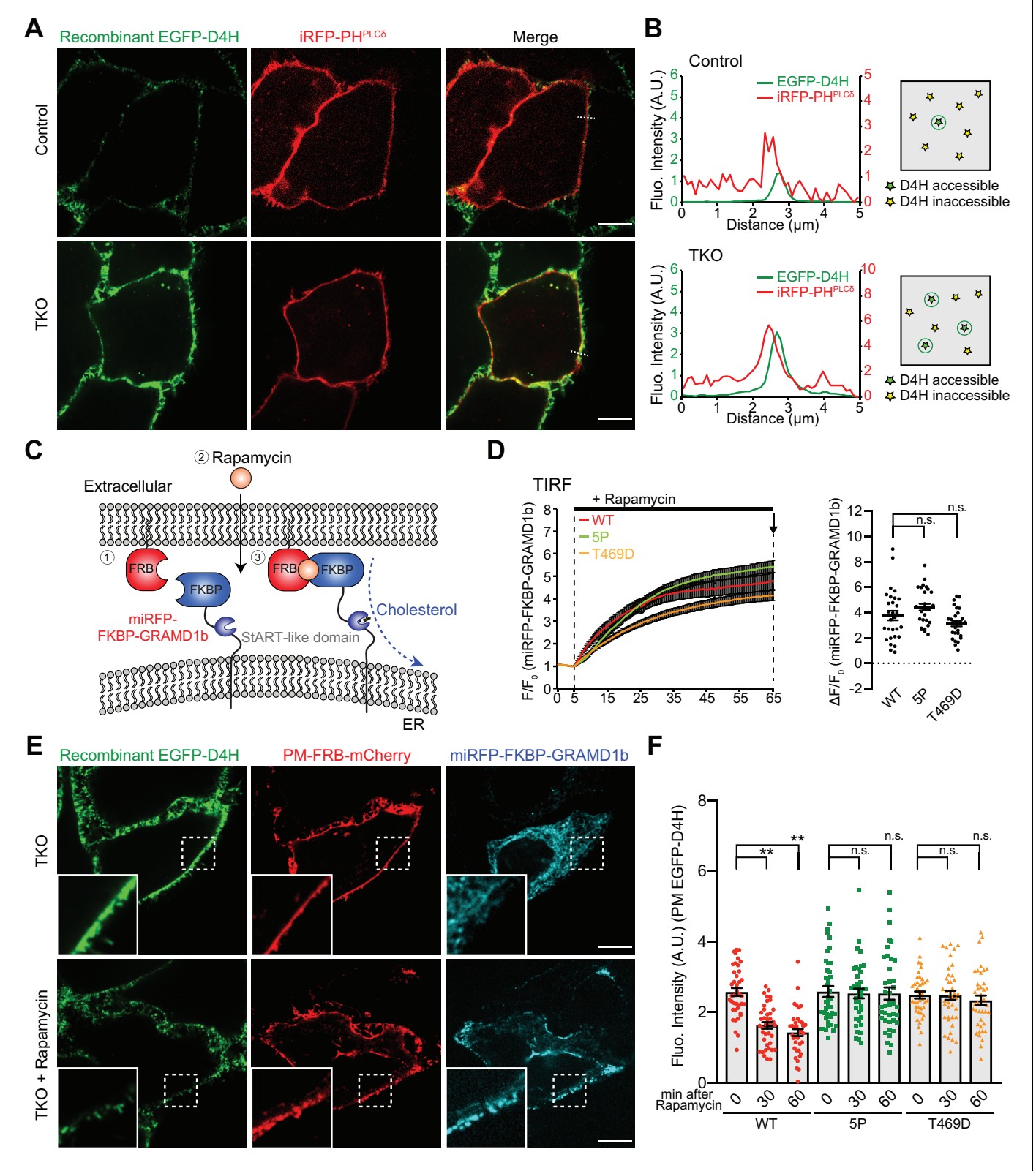

**Figure 7.** Acute recruitment of GRAMD1b to ER–PM contacts facilitates removal of the expanded pool of accessible PM cholesterol in GRAMD1 triple knockout (TKO) cells. (**A**) Left: confocal images of live wild-type (Control) and GRAMD1 TKO (TKO) HeLa cells expressing a PI(4,5)$P_2$ probe/PM marker (iRFP-PH-PLCδ) that are stained with recombinant EGFP–D4H proteins (15 µg/ml) for 15 min at room temperature. Scale bars, 10 µm. Note the increased accumulation of D4H-accessible PM cholesterol in GRAMD1 TKO cells compared to control cells, as detected by the presence of strong

*Figure 7 continued on next page*

*Figure 7 continued*

EGFP–D4H signals at the PM visualized by iRFP-PH-PLCδ. Scale bars, 10 µm. (**B**) Left: line scan analysis of the regions indicated by white dotted lines in the images shown in panel (**A**), showing the increase of EGFP–D4H signals at the PM (near the peak of iRFP-PH-PLCδ signals). Right: schematics showing the D4H-accessible pool of cholesterol on the outer leaflet of the PM (view from extracellular side) in wild-type (Control) and GRAMD1 TKO (TKO) HeLa cells. Green stars indicate D4H-accessible cholesterol, whereas yellow stars indicate D4H inaccessible cholesterol. (**C**) Schematic representation of the rapamycin-induced GRAMD1b PM recruitment strategy. GRAMD1b was rapidly recruited to the PM by rapamycin-induced dimerization of FRB and FKBP. A version of GRAMD1b with its N-terminal region, including the GRAM domain, replaced by a miRFP-tagged FKBP module (miRFP–FKBP–GRAMD1b) was expressed in GRAMD1 TKO cells together with an mCherry-tagged FRB module that is targeted to the PM (PM-FRB–mCherry). (**D**) Left: time course of normalized miRFP signal in response to rapamycin, as assessed by TIRF microscopy of GRAMD1 TKO cells expressing the indicated miRFP–FKBP–GRAMD1b constructs and PM-FRB–mCherry [wild-type (WT) and mutant versions with a StART-like domain that lacks cholesterol transport activity (5P or T469D)]. Rapamycin addition (200 nM) is indicated [mean ± SEM, n = 29 cells (WT), n = 29 cells (5P), n = 27 cells (T469D); all data are pooled from two independent experiments]. Right: values of $\Delta F/F_0$ at a time point corresponding to the end of the experiments (as shown by arrows). Dunnet's multiple comparisons test, n.s. denotes not significant. (**E**) Confocal images of GRAMD1 TKO (TKO) HeLa cells expressing miRFP–FKBP–GRAMD1b and PM-FRB–mCherry with or without rapamycin (200 nM) treatment for 60 min at 37°C and then stained with recombinant EGFP–D4H proteins (15 µg/ml) for 15 min at room temperature. Insets show at higher magnification the regions indicated by white dashed boxes. Scale bars, 10 µm. (**F**) Values of EGFP–D4H signals at the PM after background subtraction, as assessed by confocal microscopy and line scan analysis, from GRAMD1 TKO HeLa cells expressing the indicated miRFP–FKBP–GRAMD1b constructs and PM-FRB–mCherry, which were stained with recombinant EGFP–D4H protein after rapamycin addition (200 nM) for either 30 min or 60 min, as shown in panel (**E**). Peak EGFP–D4H signals around the PM marked by peak PM-FRB–mCherry signals were quantified (see Materials and methods) [mean ± SEM, n = 40 cells for each condition; all data are pooled from two independent experiments; Tukey's multiple comparisons test, **p<0.0001, n.s. denotes not significant].

The online version of this article includes the following source data and figure supplement(s) for figure 7:

**Source data 1.** Dataset for *Figure 7*.
**Figure supplement 1.** Increased D4H binding to the PM of GRAMD1 triple knockout (TKO) cells is dependent on the presence of cholesterol.
**Figure supplement 1—source data 1.** Dataset for *Figure 7—figure supplement 1*.
**Figure supplement 2.** The cholesterol-transporting property of the StART-like domain is essential for removal of the expanded pool of D4H-accessible PM cholesterol in GRAMD1 knockout cells.
**Figure supplement 2—source data 1.** Dataset for *Figure 7—figure supplement 2*.
**Figure supplement 3.** Rapamycin-induced acute recruitment of FKBP-tagged GRAMD1b to the PM in GRAMD1 triple knockout (TKO) cells.

incubated with recombinant EGFP–D4H proteins for 15 min at room temperature, washed and then imaged under spinning disc confocal microscopy.

Strikingly, rapamycin-induced PM recruitment of the chimeric GRAMD1b protein led to acute reduction in D4H-accessible PM cholesterol in GRAMD1 TKO cells within 60 min, reducing the binding of EGFP–D4H proteins to the PM (*Figure 7E,F*). Mutant versions of miRFP-FKBP–GRAMD1b carrying a StART-like domain that cannot transport cholesterol (5P and T469D mutants; see also *Figure 5B–D*, *Figure 7—figure supplement 2D–I* and *Figure 5—figure supplement 1H*) were recruited to the PM with kinetics similar those of the wild-type version (WT) (*Figure 7D* and *Figure 7—figure supplement 3B,C*; *Video 5*). However, recruitment of the mutant versions did not reduce the PM EGFP–D4H signal (*Figure 7F*), demonstrating a critical role for StART-like domain-dependent PM to ER cholesterol transport in the removal of the expanded pool of accessible PM cholesterol by GRAMD1b at ER–PM contact sites.

On the basis of these results, we conclude that GRAMD1s play a direct role in facilitating the transport of accessible PM cholesterol to the ER at ER–PM contact sites.

## Discussion

We have demonstrated that the evolutionarily conserved family of ER-anchored GRAMD1s

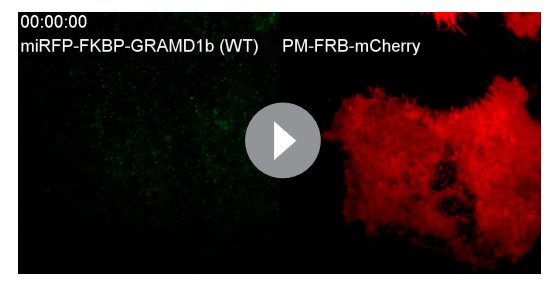

**Video 4.** Rapamycin-induced acute recruitment of GRAMD1b to the PM in GRAMD1 triple knockout (TKO) cells. GRAMD1 TKO HeLa cells expressing PM-FRB–mCherry and miRFP-FKBP–GRAMD1b (WT) were imaged under TIRF microscopy. Images were taken every 20 s, and 200 nM rapamycin was added at the 5 min time point. Note the rapamycin-induced recruitment of miRFP-FKBP–GRAMD1b to the PM. Image size, 66.1 µm x 66.1 µm.
https://elifesciences.org/articles/51401#video4

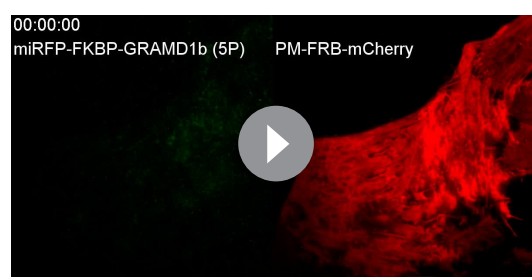

**Video 5.** Rapamycin-induced acute recruitment of a mutant version of GRAMD1b (5P) to the PM in GRAMD1 triple knockout (TKO) cells. GRAMD1 TKO HeLa cells expressing PM-FRB–mCherry and miRFP-FKBP–GRAMD1b (5P) were imaged under TIRF microscopy. Images were taken every 20 s, and 200 nM rapamycin was added at the 5 min time point. Note the rapamycin-induced recruitment of miRFP-FKBP–GRAMD1b (5P) to the PM. Image size, 66.1 µm x 66.1 µm.
https://elifesciences.org/articles/51401#video5

contribute to PM cholesterol homeostasis by sensing a transient expansion of the accessible pool of PM cholesterol and facilitating its transport to the ER at ER–PM contact sites. We have also identified the molecular mechanisms by which GRAMD1s interact with one another to form a complex, and how they are recruited to the PM. Key findings of the current study are the following:

(1) We found that GRAMD1s and GRAMD3 form homo- and heteromeric complexes and localize to discrete patches on tubular ER in mammalian cells at rest. We identified that their transmembrane domains and luminal helices, which are predicted to form amphipathic surfaces, mediated the formation of protein–protein complexes and regulated their progressive recruitment to ER–PM contacts and their functions.

(2) Using in vitro liposome sedimentation assays and live-cell imaging, we found that the GRAM domain of GRAMD1s acts as a coincidence detector, tuned to the presence of both 'unsequestered/accessible' cholesterol and anionic lipids, including phosphatidylserine, in the PM (*Figure 3I*). Importantly, the binding of the GRAM domain to membranes requires that unsequestered/accessible cholesterol exceeds a certain threshold. As the majority of cholesterol in the PM is sequestered (i.e., inaccessible), this switch-like property allows GRAMD1s to move to ER–PM contact sites only when the accessible pool of PM cholesterol transiently expands, preventing GRAMD1s from accumulating at ER–PM contacts at rest.

(3) We have deciphered the novel cellular mechanisms by which the accessibility of PM cholesterol is monitored by an LTP at ER–PM contacts. We found that GRAMD1s sense a transient expansion of the accessible pool of PM cholesterol through their GRAM domain and facilitate its transport through their StART-like domain. Disruption of their functions leads to less efficient transport of accessible PM cholesterol to the ER and reduced suppression of SREBP-2 cleavage. Importantly, we showed that the formation of GRAMD1 protein complexes, as well as the StART-like and GRAM domains of GRAMD1 proteins, is critical for the cellular functions of GRAMD1s in vivo.

(4) Our results demonstrate that removal of accessible PM cholesterol occurs within ~1 hr, when re-expressed GRAMD1 proteins are artificially recruited to ER–PM contact sites to facilitate transport of accessible cholesterol from the PM to the ER in GRAMD1 TKO cells. These experiments utilized a drug-induced dimerization approach. In addition, this GRAMD1 function requires the sterol-transporting StART-like domain. Previous studies of yeast mutants lacking Lam/Ltc proteins relied on genetic approaches (over a time scale of ~20 hr), making it difficult to interpret the significance of their lipid-transfer functions in vivo.

Using the CRISPR/Cas9 gene-editing system, we demonstrated that GRAMD1s are not essential for cell viability. Although cells that lack all three GRAMD1s grow more slowly than wild-type cells, they do not exhibit major abnormalities. Overlapping functions between mammalian GRAMD1s and other sterol-binding STARD proteins may explain the lack of major defects in these cells. Accordingly, PM lipidomics did not reveal major differences in PM cholesterol levels between GRAMD1 TKO cells and wild-type cells. Our analyses suggest, however, that GRAMD1s facilitate the transport of accessible cholesterol from the PM to the ER in response to a transient expansion of the accessible pool of PM cholesterol, thereby contributing to PM cholesterol homeostasis. First, GRAMD1 TKO cells exhibited an exaggerated accumulation of the accessible pool of PM cholesterol in response to acute hydrolysis of sphingomyelin, which liberates the sphingomyelin-sequestered pool of cholesterol into the 'accessible pool'. This was revealed by sustained binding of the PFO D4 probe to the PM and by enhanced PM recruitment of the GRAM domain of GRAMD1b, a novel biosensor for detecting acute expansion of the accessible pool of PM cholesterol that we identified in this study, in GRAMD1 TKO cells. Second, GRAMD1 TKO cells showed reduced suppression of

SREBP-2 cleavage upon hydrolysis of sphingomyelin, reflecting less efficient PM to ER transport of the newly expanded pool of accessible cholesterol in these cells. Third, GRAMD1 TKO cells exhibited a chronic expansion of the accessible pool of PM cholesterol, as detected by the D4H probe. All of these phenotypes are consistent with defects in efficient PM to ER transport of the accessible pool of cholesterol. Importantly, re-expression of GRAMD1b rescued these phenotypes, with rescue depending on the sterol-binding property of GRAMD1b's StART-like domain, the GRAM domain, and protein complex formation.

Although the transition of cholesterol between 'inaccessible' and 'accessible' pools in the PM plays crucial roles in controlling cellular cholesterol homeostasis (*Das et al., 2014*; *Endapally et al., 2019*), the molecular mechanisms by which these transitions are monitored, and the intracellular transport machinery responsible for the PM to ER transport of the accessible pool of cholesterol, have both remained elusive. Our results suggest that these two mechanisms can be coupled by non-vesicular cholesterol transport mediated by an LTP at ER–PM contact sites. We found that GRAMD1s sense a transient expansion of the accessible pool of PM cholesterol and facilitate its transport to the ER at ER–PM contact sites (*Figure 8A–C*). We found that interactions between the purified GRAM domains of GRAMD1a/b and artificial membranes that contain phosphatidylserine, which is a major acidic phospholipid in the PM, are dramatically enhanced by the presence of unsequestered/accessible cholesterol that exceeds a certain threshold. Furthermore, such interactions are modulated by sphingomyelin and phospholipids, which sequester cholesterol by forming dynamic complexes. Thus, the GRAM domain allows GRAMD1s to sense an increase in accessible PM cholesterol and facilitates the accumulation of GRAMD1s at ER–PM contacts only when the accessible pool of cholesterol transiently expands in this bilayer (*Figure 8B*). Importantly, such regulation prevents GRAMD1s from depleting PM cholesterol at steady state. Loss of GRAMD1 functions, however, leads to chronic and acute expansion of the accessible pool of PM cholesterol (*Figure 8B,C*). It has been known that accessible cholesterol, upon reaching a certain threshold, is extracted for transport to the ER and regulates cholesterol biosynthesis to maintain homeostasis. How the intracellular transport machinery senses such a sharp threshold has been unknown. The ability of the GRAMD1s to accumulate at ER–PM contacts in a switch-like fashion, by sensing a transient expansion of the accessible pool of PM cholesterol through their GRAM domain, provides a conceptual framework for this process. Importantly, GRAMD1s themselves directly facilitate the transport of the expanded pool of accessible PM cholesterol from the PM to the ER, thereby contributing to PM cholesterol homeostasis as a critical homeostatic regulator.

Levels of PM cholesterol are maintained at an equilibrium by balancing the efflux and influx of cholesterol out of and into the PM, respectively. Thus, inhibition of the efflux pathway only (e.g., by loss of GRAMD1s) disrupts this equilibrium, potentially increasing total levels of PM cholesterol. However, levels of PM cholesterol are unchanged in GRAMD1 TKO cells (*Figure 4—figure supplement 3C,D*). Thus, alternative backup systems may support the efflux of PM cholesterol (or reduce the influx of cholesterol to the PM) to maintain total cholesterol levels in the PM in the absence of GRAMD1s. For example, recent studies found that macrophages are able to dispose of accessible cholesterol by releasing PM-derived particles that are rich in cholesterol (*He et al., 2018*; *Hu et al., 2019*). It is also important to note that suppression of SREBP-2 cleavage upon sphingomyelinase treatment (an estimate of the efficiency of PM to ER transport of accessible cholesterol) is reduced but not eliminated in the absence of GRAMD1s (*Figure 6B,C*). Thus, other intracellular cholesterol transport machineries might also participate in the transport of accessible cholesterol from the PM to the ER. Such robust parallel mechanisms may be partially mediated by other non-vesicular sterol transport systems, such as those mediated by STARDs or ORPs; vesicular transport may also be involved in maintaining total levels of PM cholesterol. Further studies are needed to better understand the interplay between GRAMD1s and other sterol efflux/transfer systems in the regulation of cellular cholesterol homeostasis.

GRAMD1s interact with each other through their transmembrane domains and predicted luminal amphipathic helices (e.g., GRAMD1b homomeric and GRAMD1b/GRAMD1a heteromeric interactions). Mutant GRAMD1b proteins that lack these regions fail to interact with one another and are diffusely distributed throughout the tubular ER. Other lipid transfer proteins that are known to localize to membrane contact sites also form complexes or oligomers. These proteins include E-Syts (*Giordano et al., 2013*; *Saheki et al., 2016*), ORP2 (*Wang et al., 2019*), and ORP5/8 (*Chung et al., 2015*), as well as the yeast ERMES complex (*AhYoung et al., 2015*). ORP2 oligomerization and

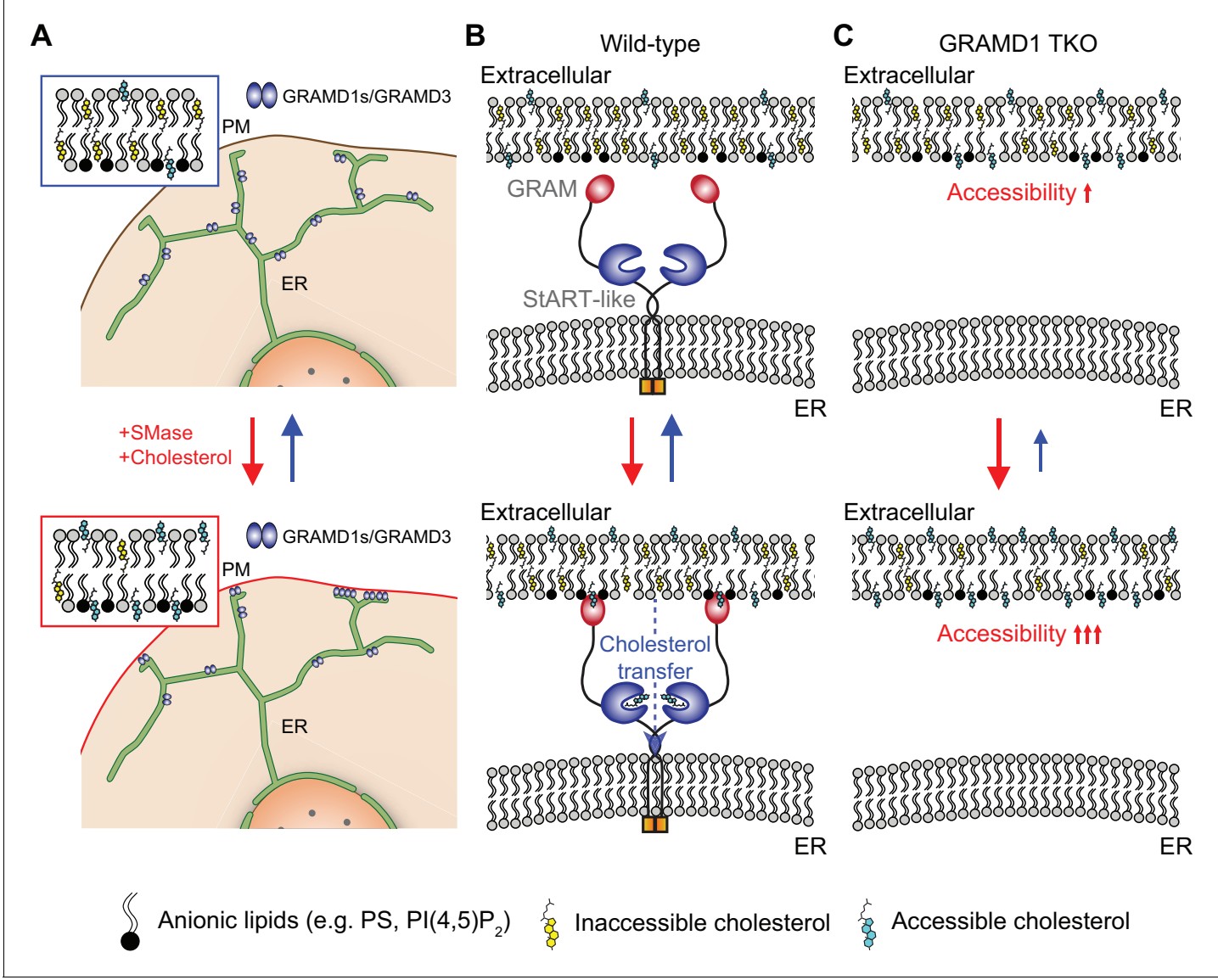

**Figure 8.** GRAMD1s facilitate the transport of the accessible pool of cholesterol from the PM to the ER, thereby contributing to PM cholesterol homeostasis. (**A**) Distinct pools of cholesterol co-exist in PM bilayers at steady state: a major pool is inaccessible (i.e. sequestered or chemically inactive) and a smaller pool is accessible (i.e. unsequestered or chemically active). Sequestration is in part mediated by sphingomyelin. Compositions of phospholipids also influence the overall accessibility of PM cholesterol. Top: at rest, GRAMD1 complexes localize on the tubular ER network with little enrichment at ER–PM contact sites. Bottom: a transient expansion of the accessible pool of PM cholesterol (e.g. hydrolysis of sphingomyelin, cholesterol loading to the PM) induces the acute recruitment of GRAMD1 complexes to ER–PM contacts once the levels of accessible PM cholesterol exceed a certain threshold. The red arrow denotes the expansion of the accessible pool; the blue arrow denotes transport of the newly expanded pool of accessible PM cholesterol to the ER by GRAMD1s and other, yet to be identified, intracellular cholesterol transport systems. (**B**) Top: in wild-type cells, the GRAM domain of GRAMD1s only weakly interacts with the PM at rest due to sequestration of the majority of cholesterol in this bilayer. Bottom: upon reaching a certain threshold of the accessibility (i.e. a transient expansion in the accessible pool of PM cholesterol), the interaction of the GRAM domain with the PM is enhanced, and GRAMD1s are recruited to ER–PM contacts. Supporting this model, the binding of the purified GRAM domains of GRAMD1a and GRAMD1b to liposomes containing phosphatidylserine, a major anionic lipid in the PM, is dramatically enhanced by the presence of unsequestered/accessible cholesterol in a dose-dependent manner with a switch-like response (*Figure 3C–F*, and *Figure 3—figure supplement 2B*). Upon recruitment of GRAMD1s to ER–PM contacts, the StART-like domain initiates the extraction of accessible PM cholesterol and facilitates its transport to the ER, contributing to the suppression of SREBP-2 cleavage and to PM cholesterol homeostasis. (**C**) Top: in GRAMD1 TKO cells at rest, D4H binding to the PM increases, indicating chronic expansion of the accessible pool of PM cholesterol due to the absence of GRAMD1s-mediated transport of accessible cholesterol from the PM to the ER (*Figure 7A,B*). Bottom: upon a transient expansion of the accessible pool of PM cholesterol, GRAMD1 TKO cells show exaggerated accumulation of the accessible pool of PM cholesterol due to the absence of GRAMD1s-mediated transport of accessible PM cholesterol to the ER. Consistent with this model, sphingomyelinase treatment leads to enhanced recruitment of the GRAM

*Figure 8 continued on next page*

*Figure 8 continued*

domain of GRAMD1b, a novel biosensor for detecting acute expansion of the accessible pool of PM cholesterol that we identified in this study, to the PM in GRAMD1 TKO cells (*Figure 4D,E*). Reduced transport of accessible cholesterol from the PM to the ER in GRAMD1 TKO cells also results in less efficient suppression of SREBP-2 cleavage and continuous increase in the binding of recombinant D4 proteins to the PM (*Figure 6B,C* and *Figure 6— figure supplement 2A,B*). Other parallel cholesterol transport and regulatory systems, yet to be identified, may operate to maintain the total levels of PM cholesterol in the absence of GRAMD1s.

ERMES assembly enhance the ability of these proteins to transfer lipids (*Kawano et al., 2018*; *Wang et al., 2019*). Accordingly, mutant GRAMD1b proteins that lack the ability to form protein complexes accumulate less robustly at ER–PM contacts, and less effectively rescue phenotypes associated with GRAMD1 TKO cells. Thus, the formation of GRAMD1 complexes plays important roles in regulating GRAMD1 localization and function.

A recent study by *Sandhu et al. (2018)* reported that GRAMD1s facilitate the transport of PM cholesterol that is additionally loaded from external sources. In particular, they showed that GRAMD1b mediates PM to ER transport of HDL-derived cholesterol that is taken up by adrenal glands via the scavenger receptor SR-B1 in mice. On the basis of these results, they proposed that GRAMD1s might also play more general roles in intracellular cholesterol transport by sensing accessible cholesterol in the PM. Our findings support this notion and further demonstrate that GRAMD1s contribute to PM cholesterol homeostasis and cholesterol metabolism by sensing a transient expansion of the accessible pool of PM cholesterol by their GRAM domain, and facilitate the transport of this cholesterol to the ER by their StART-like domain. Interestingly, GRAMD1a is broadly expressed in many tissues with particular enrichment in the brain. Future studies will be needed to determine the physiological functions of GRAMD1s in other tissues in mammals.

In summary, our study has demonstrated that ER-localized GRAMD1s help to maintain PM cholesterol homeostasis in trans via their ability to transport accessible cholesterol from the PM to the ER at ER–PM contact sites. Our results show that GRAMD1s sense a transient expansion of the pool of accessible PM cholesterol and facilitate its transport to the ER through non-vesicular transport. These proteins probably perform additional functions, as yeast mutants that lack Lam/Ltc proteins show pleiotropic defects in cell signaling, including altered mTOR kinase signaling (*Murley et al., 2017*). Furthermore, the StART-like domain of GRAMD1b has been shown to transport $PI(4,5)P_2$ in addition to cholesterol (*Horenkamp et al., 2018*). Thus, potential changes in the properties of other organelles and the PM in GRAMD1 TKO cells deserve future investigations. Further elucidating the physiological functions of these proteins, as well as other sterol-transfer proteins, at membrane contact sites will be important to gain insight into how cellular cholesterol homeostasis is regulated.

# Materials and methods

## Antibodies and chemicals
Primary and secondary antibodies, chemicals, lipids and other reagents used in this study are listed in *Supplementary file 1*.

## DNA plasmids
DNA plasmids used in this study are listed in *Supplementary file 1*; the sequences of oligos and primers used are listed in *Supplementary file 2*.

### For mammalian expression
Cloning of EGFP–GRAMD1a, EGFP–GRAMD1b, EGFP–GRAMD1c and EGFP–GRAMD3
cDNAs of GRAMD1a/Aster-A (NP_065946.2), GRAMD1b/Aster-B (NP_065767.1), GRAMD1c/Aster-C (NP_060047.3), and GRAMD3 (NP_001139791.1) were amplified by PCR using the following primer sets: GRAMD1a — GRAMD1a_F and GRAMD1a _Stop_R; GRAMD1b — GRAMD1b_F and GRAMD1b_Stop_R; GRAMD1c — GRAMD1c_F and GRAMD1c_Stop_R; and GRAMD3 — GRAMD3_F and GRAMD3_Stop_R. PCR products were ligated at XhoI and KpnI sites for GRAMD1a, GRAMD1b, and GRAMD1c, and at SalI and KpnI sites for GRAMD3 in the pEGFP-C1 vector.

## Cloning of Myc–GRAMD1a, Myc–GRAMD1b, mRuby–GRAMD1b, Myc–GRAMD3, and mCherry –GRAMD3

cDNAs corresponding to GRAMD1a of EGFP GRAMD1a and GRAMD1b of EGFP–GRAMD1b were excised and ligated into pMyc-C1 and pmRuby-C1 vectors in the XhoI and KpnI sites to generate Myc–GRAMD1a, Myc–GRAMD1b, and mRuby–GRAMD1b, respectively.

cDNA corresponding to GRAMD3 of EGFP–GRAMD3 was excised and ligated into pMyc-C1 and pmCherry-C1 vectors in the SalI and KpnI sites to generate Myc–GRAMD3 and mCherry–GRAMD3, respectively.

Cloning of EGFP–GRAMD1b ΔGRAM and EGFP–GRAMD1b ΔHelix cDNAs corresponding to the residues 164–738 (ΔGRAM) and 1–688 (ΔHelix) of GRAMD1b were PCR amplified and ligated into pEGFP-C1 vector in the XhoI and KpnI sites, using the following primer sets: GRAMD1b DeltaNterm F and GRAMD1b_Stop_R for generating EGFP–GRAMD1b (164-738) (ΔGRAM); GRAMD1b_F and 3'_KpnI-stop_GRAMD1b_683aa for generating EGFP-GRAMD1b (ΔHelix).

## Cloning of EGFP–GRAMD1b (4E) and EGFP–GRAMD1b (5E)

Hydrophobic amino-acid residues (W678, L681, L682 and Y688) present in the luminal region of GRAMD1b were mutated to glutamic acid using site-directed mutagenesis in EGFP–GRAMD1b with the primer set, GRAMD1b_Helix4E_F and GRAMD1b_Helix4E_R, to generate EGFP–GRAMD1b (4E).

Hydrophobic amino-acid residues (L693, W696, I699, I700, and L707) present in the predicted luminal amphipathic helix of GRAMD1b were mutated to glutamic acid using site-directed mutagenesis in EGFP–GRAMD1b with the primer set, GRAMD1b_Helix_2_5E_F and GRAMD1b_Helix_2_5E_R, to generate EGFP–GRAMD1b (5E).

## Cloning of mRuby–GRAMD1b (T469D) and mRuby–GRAMD1b (Y430A, V445A)

The amino-acid residue present in the StART-like domain that was reported to be important for transporting cholesterol (T469) (*Horenkamp et al., 2018*) was mutated to aspartic acid using site-directed mutagenesis in mRuby–GRAMD1b with the primer set, GRAMD1b_T469D_F and GRAMD1b_T469D_R, to generate mRuby–GRAMD1b (T469D).

The amino-acid residues present in the StART-like domain that were predicted to be critical for binding cholesterol on the basis of our structural modeling (Y430 and V445) were mutated to alanine using site-directed mutagenesis in mRuby–GRAMD1b, with the primer set, GRAMD1b_Y430A_-V445A_F and GRAMD1b_Y430A_V445A_R, to generate mRuby-GRAMD1b (Y430A, V445A).

## Cloning of mRuby–GRAMD1b (5P)

Four residues (L434, T435, N436, and L438) on the loop of the StART-like domain of GRAMD1b were mutated to proline using site-directed mutagenesis in mRuby–GRAMD1b using the primer set, GRAMD1b_434LTNPL_F and GRAMD1b_434LTNPL_R, to generate mRuby-GRAMD1b (5P).

## Cloning of miRFP-FKBP–GRAMD1b (WT), miRFP-FKBP–GRAMD1b (5P) and miRFP-FKBP–GRAMD1b (T469D)

The plasmid miRFP-FKBP–GRAMD1b (WT) was generated using mCherry–pMag(x3)-MTMR1 (*Benedetti et al., 2018*) as a backbone. cDNA corresponding to the residues 164–738 (ΔGRAM) of GRAMD1b was amplified by PCR using the primer set, 5'-KpnI-GRAMD1b-164aa-S and 3'-BamHI-stop-GRAMD1b-C-AS, and ligated at the KpnI and BamHI sites of mCherry–pMag(x3)-MTMR1 vector. Subsequently, cDNA corresponding to miRFP of LAMP1–miRFP was amplified by PCR using the primer set, 5'_NheI_miRFP_S and 3'_NotI_miRFP_AS, and ligated at the NheI and NotI sites of the vector to generate miRFP-pMag(x3)–GRAMD1b (164-738). Finally, cDNA corresponding to a FKBP module of mCherry–FKBP-MTM1 was amplified by PCR using the primer set, 5'_NotI_FKBP_linker_S and 3'_KpnI_FKBP_linker_AS, and ligated at the NotI and KpnI sites of miRFP-pMag(x3)–GRAMD1b (164-738) to generate miRFP-FKBP–GRAMD1b (WT).

cDNA corresponding to GRAMD1b (164-738) of miRFP-FKBP–GRAMD1b (WT) was replaced with cDNA corresponding to either GRAMD1b (164–738:5P) or GRAMD1b (164–738:T469D) by digesting either mRuby–GRAMD1b (5P) or mRuby–GRAMD1b (T469D), respectively, and ligating the resulting

fragments at the EcoRV and MluI sites of miRFP-FKBP–GRAMD1b (WT) to generate miRFP-FKBP–GRAMD1b (5P) and miRFP-FKBP–GRAMD1b (T469D).

Cloning of EGFP–GRAMD1a GRAM (EGFP–GRAM$_{1a}$), EGFP–GRAMD1b GRAM (EGFP–GRAM$_{1b}$), and EGFP–GRAMD1c GRAM (EGFP–GRAM$_{1c}$) cDNAs corresponding to the GRAM-domain residues 81–220 of GRAMD1a, 92–207 of GRAMD1b, and 65–186 of GRAMD1c were PCR amplified and ligated into pEGFP-C1 vector at the XhoI and KpnI sites, using the following primer sets: GRAMD1a_GRAM Domain_F and GRAMD1a_GRAM Domain_R for generating EGFP–GRAMD1a GRAM; GRAMD1b_Short GRAM Domain_F and GRAMD1b_Short GRAM Domain_R for generating EGFP–GRAMD1b GRAM; and GRAMD1c_GRAM Domain_F and GRAMD1c_GRAM Domain_R for generating EGFP–GRAMD1c GRAM.

Cloning of EGFP–GRAMD1b (TM swap) and mRuby–GRAMD1b (TM swap) cDNAs corresponding to the residues 404–621 of GRAMD1b and 67–96 of Sec61β were individually amplified by PCR with the following primer sets: 5'_GRAMD1b_iEcoRV_Fw_HiFi and 3'_GRAMD1b_621_Rv_HiFi for GRAMD1b; and 5'_GRAMD1b-Sec61b_TM_Fw_HiFi and 3'_pEGFP_Sec61b_TM_Rv_HiFI for Sec61β. The two PCR products were simultaneously ligated at the EcoRV and KpnI sites of mRuby–GRAMD1b by DNA HiFi assembly kit (NEB) to generate mRuby–GRAMD1b (TM swap). cDNA corresponding to GRAMD1b (TM swap) was excised and ligated into pEGFP-C1 vector at the XhoI and KpnI sites to generate EGFP–GRAMD1b (TM swap).

## Cloning of mRuby–OSBP and mRuby–ORP9

cDNAs of OSBP (BC011581) and ORP9 (BC025978) were amplified by PCR and ligated at HindIII and BamHI sites and XhoI and HindIII sites, respectively, in the pmRuby-C1 vector, using the following primer sets (5'_HindIII_OSBP_NS and 3'_BamHI_stop_OSBP_CAS for generating mRuby–OSBP; and 5'_XhoI_ORP9_NS and 3'_HindIII_stop_ORP9_CAS for generating mRuby-ORP9).

## Cloning of mRuby–ORP4 and mCherry–STARD4

gBlocks (IDT) containing cDNA of ORP4 (BC118914) and STARD4 (BC042956) were synthesized (ORP4_Frag_1_XhoI, ORP4_Frag_2_BamHI,and XhoI_STARD4_KpnI) and ligated at the XhoI and BamHI sites of pmRuby-C1 vector to generate mRuby-ORP4 and at the XhoI and KpnI sites of the pmCherry-C1 vector to generate mCherry–STARD4, respectively, using a DNA HiFi assembly kit (NEB).

A clone of PM-FRB–mCherry cDNA corresponding to the PM targeting signal (the residues 1–20 of mouse GAP43) and the FRB module was digested from PM-FRB-CFP (a gift from the De Camilli Lab) and ligated into the NheI and AgeI sites of the pmCherry-N1 vector.

## Recombinant protein purification

### Cloning of the StART-like and GRAM domains of human GRAMD1s

The cDNAs corresponding to the StART-like and GRAM domains of human GRAMD1 proteins were ligated into the pNIC28-Bsa4 vector with an N-terminal His$_6$-tag and a TEV-protease cleavage site (residues 366–537 for GRAMD1a$_{StART}$; 375–545 for GRAMD1b$_{StART}$; 325–500 for GRAMD1c$_{StART}$; 81–220 for GRAM$_{1a}$; and 70–231 for GRAM$_{1b}$) via ligation-independent cloning to generate pNIC28-Bsa4 GRAMD1a StART L366-S537, pNIC28-Bsa4 GRAMD1b StART Q375-E545, pNIC28-Bsa4 GRAMD1c StART L325-I500, pNIC28-Bsa4 GRAMD1a GRAM 81–220 and pNIC28-Bsa4 GRAMD1b GRAM 70–231, respectively.

### Cloning of GRAMD1a StART-like domain mutant (5P)

Four residues (I429, S430, N431, L433) on the loop of the StART-like domain of GRAMD1a were mutated to proline using site-directed mutagenesis in pNIC28-Bsa4 GRAMD1a StART L366-S537 using the primer set, GRAMD1a_I429SNPL_F and GRAMD1a_I429SNPL_R, to generate pNIC28-Bsa4 GRAMD1a StART L366-S537 5P.

### Cloning of the GRAMD1b StART-like domain mutant (5P)

Four residues (L434, T435, N436, and L438) on the loop of the StART-like domain of GRAMD1b were mutated to proline using site-directed mutagenesis in pNIC28-Bsa4 GRAMD1b StART Q375-

E545 using the primer set, GRAMD1b_434LTNPL_F and GRAMD1b_434LTNPL_R, to generate pNIC28-Bsa4 GRAMD1b StART Q375-E545 5P.

## Cloning of the GRAMD1b StART-like domain mutant (T469D)

T469 was mutated to aspartate using site-directed mutagenesis in pNIC28-Bsa4 GRAMD1b StART Q375-E545 using the primer set, GRAMD1b_T469D_F and GRAMD1b_T469D_R, to generate pNIC28-Bsa4 GRAMD1b StART Q375–E545 T469D.

Cloning of EGFP-linker-GRAMD1b luminal helix and EGFP-linker-GRAMD1b luminal helix 5E gBlocks (IDT) containing EGFP–(GGGS)$_3$–GRAMD1b luminal helix (674–718) and EGFP–(GGGS)$_3$–GRAMD1b luminal helix (674–718), which carry 5E mutations, were synthesized (EGFP–GRAMD1b 674–718aa and EGFP–GRAMD1b_Helix_5E) and amplified by PCR using the primer set, 5'NcoI_eGFP_1b_Helix and 3'XhoI_eGFP_1b_Helix. The PCR products were then ligated at NcoI and XhoI sites in the pET28b(+) vector to generate EGFP-linker-luminal He and EGFP-linker-luminal He with 5E.

Clones of EGFP–D4 and EGFP–D4H (D434S) gBlock (IDT) containing EGFP-D4 were synthesized (BsrG1-D4_E.coli-BamHI) and ligated into the pNIC28-Bsa4 vector via ligation-independent cloning to generate EGFP–D4–CLOPF-ec01. D434 was mutated to serine (*Maekawa and Fairn, 2015*) using site-directed mutagenesis in EGFP–D4–CLOPF-ec01 using the primer set, D4_D434S_F and D4_D434S_R, to generate pNIC28-Bsa4 EGFP–D4H.

## Cell culture and transfection

HeLa and COS-7 cells were cultured in Dulbecco's modified Eagle's medium (DMEM) containing 10% or 20% fetal bovine serum (FBS) and 1% penicillin/streptomycin at 37°C and 5% CO$_2$. Transfection of plasmids was carried out with Lipofectamine 2000 (Thermo Fisher Scientific). Both wild-type and genome-edited HeLa cell lines were routinely verified as free of mycoplasma contamination at least every two months, using MycoGuard Mycoplasma PCR Detection Kit (Genecopoeia). No cell lines used in this study were found in the database of commonly misidentified cell lines that is maintained by ICLAC and NCBI Biosample.

## Fluorescence microscopy

For imaging experiments, cells were plated onto 35 mm glass bottom dishes at low density (MatTek Corporation). All live-cell imaging was carried out one day after transfection.

Spinning disc confocal (SDC) microscopy (*Figures 3A*, *4D*, *7A,E*, *Figure 4—figure supplement 2A*, *Figure 4—figure supplement 3A*, *Figure 5—figure supplement 2A*, *Figure 7—figure supplement 1A*, *Figure 7—figure supplement 2A,E,G*) and super-resolution SDC-structured illumination microscopy (SDC-SIM) (*Figures 1B,C*, *2C,H–I* and *Figure 1—figure supplement 1A,B*) were performed on a setup built around a Nikon Ti2 inverted microscope equipped with a Yokogawa CSU-W1 confocal spinning head, a Plan-Apo objective (100 × 1.45 NA), a back-illuminated sCMOS camera (Prime 95B; Photometrics), and a super-resolution module (Live-SR; Gataca Systems) that was based on structured illumination with optical reassignment and image processing (*Roth and Heintzmann, 2016*). The method, known as multifocal structured illumination microscopy (*York et al., 2012*), makes it possible to double the resolution and the optical sectioning capability of confocal microscopy simultaneously. The maximum resolution is 128 nm with a pixel size in super-resolution mode of 64 nm. Excitation light was provided by 488 nm/150 mW (Coherent) (for GFP), 561 nm/100 mW (Coherent) (for mCherry/mRFP/mRuby) and 642 nm/110 mW (Vortran) (for iRFP/miRFP) (power measured at optical fiber end) DPSS laser combiner (iLAS system; Gataca systems). All image acquisition and processing was controlled by MetaMorph (Molecular Device) software. Images were acquired with exposure times in the 400–500 msec range.

Total internal reflection fluorescence (TIRF) microscopy (*Figures 3B,G,H*, *4E*, *5E,F*, *6A,F,G*, *7D* and *Figure 3—figure supplement 1A*, *Figure 4—figure supplement 2B*, *Figure 5—figure supplement 2B-E*, *Figure 6—figure supplement 1C,D*, *Figure 7—figure supplement 3A-C*) was performed on a setup built around a Nikon Ti2 inverted microscope equipped with a HP Apo-TIRF objective (100 × 1.49 NA), and a back-illuminated sCMOS camera (Prime 95B; Photometrics). Excitation light was provided by 445 nm/25 mW (for CFP), 488 nm/70 mW (for GFP), 561 nm/70 mW (for mCherry/mRFP/mRuby) and 647 nm/125 mW (for iRFP/miRFP) (power measured at optical fiber end)

DPSS laser combiner (Nikon LU-NV laser unit), coupled to the motorized TIRF illuminator through an optical fiber cable. Critical angle was maintained at different wavelengths throughout the experiment from the motorized TIRF illuminator. Acquisition was controlled by Nikon NIS-Element software. For time-lapse imaging, images were sampled at 0.05 Hz with exposure times in the 200–500 msec range.

Cells were washed twice and incubated with $Ca^{2+}$ containing buffer (140 mM NaCl, 5 mM KCl, 1 mM $MgCl_2$, 10 mM HEPES, 10 mM glucose, and 2 mM $CaCl_2$ [pH 7.4]) before imaging with either an SDC microscope or a TIRF microscope. All types of microscopy were carried out at 37°C except for the experiments with recombinant EGFP–D4H proteins, which were performed at room temperature via SDC microscopy.

## Drug stimulation for time-lapse TIRF imaging

For all time-lapse TIRF imaging experiments with drug stimulation, drugs were added to the cells 5 min after the initiation of the imaging, except for methyl-β-cyclodextrin (MCD) treatment, where 10 mM MCD (Sigma-Aldrich/Merck) was added to the cells as indicated in a figure (*Figure 4—figure supplement 2B*). Other drugs were used at the following concentrations: 200 μM cholesterol/MCD complex generated as described previously (*Brown et al., 2002*); 100 mU/ml sphingomyelinase (SMase) (Sigma-Aldrich/Merck); and 200 nM rapamycin (Sigma-Aldrich/Merck).

## Assays with recombinant EGFP–D4H proteins

For the assays with recombinant EGFP–D4H proteins (*Figure 7A,B,E,F* and *Figure 7—figure supplement 1A,B*, *Figure 7—figure supplement 2A,B,E,F,G,H*), transfected cells with or without drug treatment were washed once with $Ca^{2+}$ containing buffer and subsequently incubated with the same buffer containing recombinant EGFP-D4H proteins (15 μg/ml) for 15 min at room temperature. Cells were then washed twice with the same buffer without EGFP-D4H proteins and immediately imaged under SDC microscopy at room temperature.

For the experiments with MCD treatment (*Figure 7—figure supplement 1A,B*), GRAMD1 TKO cells were transfected with iRFP-PH-PLCδ. On the following day, cells were washed with $Ca^{2+}$-containing buffer and incubated for 30 min at room temperature with the same buffer containing 10 mM MCD (Sigma-Aldrich/Merck) before staining with recombinant EGFP–D4H proteins for SDC microscopy.

For rescue experiments with mRuby–GRAMD1 constructs (*Figure 7—figure supplement 2A,B,E, F,G,H*), GRAMD1 TKO cells were transfected with iRFP-PH-PLCδ together with the indicated mRuby–GRAMD1 constructs. On the following day, cells were stained with recombinant EGFP–D4H proteins and imaged under SDC microscopy.

For rapamycin-induced dimerization experiments (*Figure 7E,F*), GRAMD1 TKO cells were transfected with PM-FRB–mCherry and one of the three miRFP-FKBP–GRAMD1b constructs (WT, 5P, or T469D). On the following day, 200 nM rapamycin (Sigma-Aldrich/Merck) was added to the culture media, which was further incubated for either 30 min or 60 min at 37°C before staining with recombinant EGFP–D4H proteins for SDC microscopy.

## Image analysis

All images were analyzed off-line using Fiji (http://fiji.sc/wiki/index.php/Fiji). Quantification of fluorescence signals was performed using Excel (Microsoft) and Prism 7 or 8 (GraphPad Software). All data are presented as mean ± SEM. In dot plots, each dot represents the value from a single cell with the black bar as the mean.

For time-lapse imaging via TIRF microscopy, changes in PM fluorescence over time were analyzed by manually selecting regions of interest covering the largest possible area of the cell foot-print. Mean fluorescence intensity values of the selected regions were obtained and normalized to the average fluorescence intensity before stimulation after background subtraction.

For analysis of the binding of recombinant EGFP–D4H proteins to the PM via SDC microscopy, line scan analysis was performed. A line of 5 μm in length was manually drawn around the PM (see dashed white lines in *Figure 7A* and *Figure 7—figure supplement 1A*), and EGFP fluorescence intensity along the manually drawn line was measured. The minimum fluorescence intensity along

the line was subtracted from the maximum fluorescence intensity (corresponding to the PM-bound EGFP–D4H fluorescence) and plotted for quantification.

## Generation of GRAMD1 knockout HeLa cell lines

The GRAMD1B, GRAMD1A and GRAMD1C genes were sequentially targeted to generate GRAMD1 triple knockout cells. The sequences of oligos and primers used are listed in *Supplementary file 2*.

For the generation of HeLa cells lacking GRAMD1b, control wild-type HeLa cells were transfected with a plasmid encoding spCas9 and the GRAMD1b-targeting guide RNA (*Figure 4A*), followed by isolation of individual clones by dilution cloning. Two clones (#10 and #17) were further characterized by sequencing and immunoblotting (i.e. western blotting). These analyses revealed deletions and insertions within the guide RNA-binding sites, frame-shift and early termination in the open-reading frame of GRAMD1B gene, and the loss of GRAMD1b protein expression (*Figure 4—figure supplement 1A,C*). To generate GRAMD1a/1b double knockout (DKO) cell lines, a subclone of the GRAMD1b KO cell line #10 was transfected with a plasmid encoding spCas9 and the GRAMD1a-targeting guide RNA with ssDNA oligos containing stop codons and homology-arms (*Figure 4A*). These cells were subjected to single cell sorting, and individually isolated clones [lines #38 and #40 (hereafter GRAMD1a/1b DKO #38 and GRAMD1a/1b DKO #40)] showed insertion of ssDNA within the guide RNA-targeted locus, resulting in the lack of GRAMD1a protein expression (*Figure 4B* and *Figure 4—figure supplement 1B,D*).

To generate GRAMD1 triple knockout (TKO) cell lines, the GRAMD1a/1b DKO #40 cell line was transfected with two plasmids encoding spCas9 and each one of the two GRAMD1b-targeting guide RNAs (*Figure 4A*). Two clones (GRAMD1a/1b/1c TKO #1 and GRAMD1a/1b/1c TKO #15) that showed large deletions in the exon 11 of GRAMD1C, as assessed by genomic PCR (*Figure 4—figure supplement 1E*), were isolated and knock-outs were confirmed by direct sequencing (*Figure 4C*). In all figures and texts, TKO denotes GRAMD1a/1b/1c TKO #15 unless stated.

### GRAMD1b knockout

The genomic sequence surrounding the exon 13, which encodes the amino-acid stretch in the StART-like domain of human GRAMD1b, was analyzed for potential CRISPR/Cas9 targets in silico using the Cas9 design target tool (http://crispr.mit.edu) (*Hsu et al., 2013*). The GRAMD1B genomic sequence targeted by the predicted CRISPR gRNA is: TCGCTACACGCTCACCCGTGTGG (GRAMD1b-sgRNA).

The CRISPR targeting site was synthesized by annealing GRAMD1b-sgRNA#1_S and GRAMD1b-sgRNA#1_AS and sub-cloned into a human codon-optimized Cas9 and chimeric gRNA expression plasmid that carries puromycin resistance, pSpCas9(BB)−2A-Puro (PX459), obtained from Addgene (Plasmid 48139) (*Ran et al., 2013*) to generate PX459-GRAMD1B_Back.

HeLa cells were transiently transfected with the PX459-GRAMD1B_Back plasmid. 24 hr after transfection, cells were supplemented with growth medium containing puromycin (1.5 µg/mL) and incubated for 72 hr. Cells that were resistant to puromycin selection were then incubated with puromycin-free medium for 24 hr before harvesting for dilution cloning, and then assessed by genotyping PCR using the primer set, Genotyping_1B_F2 and Genotyping_1B_R1, to obtain GRAMD1b knockout cells.

### GRAMD1a knockout

The genomic sequence surrounding the exon 13, which encodes the amino-acid stretch in the StART-like domain of human GRAMD1a, was analyzed in silico using the Cas9 design target tool (http://crispr.mit.edu) (*Hsu et al., 2013*). The GRAMD1A genomic sequence targeted by the predicted CRISPR gRNA is: GGACTCCGAGGTGCTGACGCAGGG (GRAMD1a-sgRNA).

The CRISPR targeting site was synthesized by annealing GRAMD1a_sgRAN#1_S and GRAMD1a_sgRNA#1_AS and sub-cloned into PX459 (*Ran et al., 2013*) to generate PX459-GRAMD1A_V2_Front.

To knock-in the sequence with stop codons, ssDNA containing stop codons and homology-arms surrounding the guide RNA targeting site was designed. The ssDNA of the reverse complementary sequence was synthesized by IDT and used for the transfection with the PX459-GRAMD1A_V2_Front plasmid. The sequence of ssDNA was: <u>GTGGGCAGTGTAGAAGTAGTCCTGGTAGGGGATGCCC</u>

<u>TGC</u>GGATCCCGGGCCCGCGGTACCGAATTCGAAGCTTGAGCTCGAGATCTActagttaatca<u>GTCAG-
CACCTCGGAGTCCACCACACACCCGCCGGCCTGGG</u>, where homology arms are indicated by the
underline (ssDNA_StopKI_HR_GRAMD1a_V2).

Control HeLa cells and GRAMD1b knockout cell line #10 were transiently transfected with the
PX459-GRAMD1A_V2_Front plasmid with ssDNA. 24 hr after transfection, cells were supplemented
with growth medium containing puromycin (1.5 μg/mL) and incubated for 72 hr. Cells
that were resistant to puromycin selection were then incubated with puromycin-free medium for 24
hr before harvesting for single-cell sorting, and individually isolated cell clones were assessed by
genotyping PCR using the primer set, GRAMD1a_Genotyping_V2V3_F2 and GRAMD1a_Genoty-
ping_V2V3_R2, to obtain GRAMD1a knockout and GRAMD1a/1b DKO cell lines.

### GRAMD1c knockout

The genomic sequence surrounding the exon 11, which encodes the amino-acid stretch in the
StART-like domain of human GRAMD1c, was analyzed in silico using the Cas9 design target tool
(http://crispr.mit.edu) (*Hsu et al., 2013*). The GRAMD1C genomic sequences targeted by the pre-
dicted CRISPR gRNAs are: TAGATGGTAGTATCTACCCCTTGG (GRAMD1c-sgRNA#1) and ACTA
TTAAGGACTATAGTGTAGG (GRAMD1c-sgRNA#2).

The two CRISPR targeting sites were synthesized by annealing GRAMD1c-sgRNA#1_S and
GRAMD1c-sgRNA#1_AS for GRAMD1c-sgRNA#1, and GRAMD1c-sgRNA#2_S and GRAMD1c-
sgRNA#2_AS for GRAMD1c-sgRNA#2, respectively. These sites were then individually sub-cloned
into PX459 (*Ran et al., 2013*) to generate PX459-GRAMD1c_sgRNA_#1 and PX459-
GRAMD1c_sgRNA_#2.

GRAMD1a/1b DKO cell line #40 was transiently transfected with the two GRAMD1c CRISPR/Cas9
plasmids, PX459-GRAMD1c_sgRNA_#1 and PX459-GRAMD1c_sgRNA_#2. 24 hr after transfection,
cells were supplemented with growth medium containing puromycin (1.5 μg/mL) and incubated for
72 hr. Cells that were resistant to puromycin selection were then incubated with puromycin-free
medium for 24 hr before harvesting for single-cell sorting, and individually isolated clones were
assessed by genotyping PCR using the primer set, GRAMD1c_Genotyping_F1 and GRAMD1c_Ge-
notyping_R1, to obtain GRAMD1a/1b/1c triple knockout cell lines.

### Sequencing of mutant alleles

For GRAMD1a and GRAMD1b knockout cells, sequencing of mutated alleles was carried out by
cloning PCR products into the pCR4 Blunt-TOPO vector using the Zero Blunt TOPO PCR Cloning Kit
for sequencing (Thermo Fisher Scientific). Biallelic insertions/deletions were confirmed by sequenc-
ing at least 10 individual colonies. The same primers were used as genotyping primers. For
GRAMD1c knockout cells, sequencing of mutated alleles was carried out by direct-sequencing of the
genomic PCR products. The same primers were used as genotyping primers.

## Biochemical analyses

### Plasma membrane isolation and protein extraction

The procedure was modified from *Cohen et al. (1977)* and *Saheki et al. (2016)*. Briefly, 2 g of Cyto-
dex three microcarrier beads (Sigma-Aldrich/Merck) were reconstituted in 100 ml phosphate-buff-
ered saline (PBS), autoclaved and coated by incubation with a poly-D-lysine solution overnight at 37°
C. Cells were added to the reconstituted beads in sterile PETG flasks (Thermo Fisher Scientific),
allowed to attach to the beads for 4 hr with a gentle stirring every 30 min, and then further incu-
bated overnight with continuous stirring on a rotating incubator at 37°C with 5% $CO_2$. Beads were
subsequently collected by spontaneous sedimentation and incubated with 220 mM sucrose and 40
mM sodium acetate (pH 5.0) for 5 min at room temperature (*Cohen et al., 1977*). After the acid
treatment, beads were collected, incubated with a hypotonic solution [10 mM Tris-HCl (pH 8.0)] and
vortexed for 10 s. A 10 s sonication pulse was then applied to the beads with Vibra Cell VCX130
(Sonics and Materials, Inc) in the same solution. Beads were finally washed three times with the same
solution and once with PBS for lipidomic analysis and for protein extraction with SDS lysis buffer [10
mM Tris-HCl, 150 mM NaCl, 2% SDS (pH 8.0)].

## Lipidomics

Lipids were extracted from either total cells still attached to the beads (total lipids), or from beads-bound plasma membranes prepared as described above (PM lipids), according to the Bligh-Dyer method (*Bligh and Dyer, 1959*).

Samples were then spiked with a SPLASH LIPIDOMIX deuterated lipid internal standard solution (Avanti Polar Lipids) and lipidomic analyses were performed in shotgun mode, using a Nanomate nanoflow electrospray infusion device (Advion BioSciences) coupled to a QExactive mass spectrometer (Thermo Fisher Scientific) in both positive and negative polarity (*Surma et al., 2015*). For measurements of cholesterol, samples were spiked with cholesterol D6, derivatized with acetyl chloride (*Lyons et al., 1981*) and analyzed in direct infusion positive ion mode. Lipids were identified with LipidXplorer (*Sales et al., 2017*), using the accurate mass of intact lipid ions for identification and characteristic masses of fragment ions for confirmation. Quantification of lipids was performed using class-specific internal standards. To ensure analytical quality, WT/KO samples, as well as PM/total lipids, were injected in alternating order and bracketed by replicate injections of a pooled quality control (QC) sample and solvent blanks. Only analytes that showed RSD <20% in the QC samples, and a blank/sample ratio of <10%, were reported.

## Immunoblotting and immunoprecipitation

### Immunoblotting

HeLa cells were lysed in buffer containing 2% SDS, 150 mM NaCl, 10 mM Tris (pH 8.0), and incubated at 60°C for 20 min followed by additional incubation at 70°C for 10 min. The lysates were treated with benzonase nuclease (Sigma-Aldrich/Merck or SantaCruz) for 10–15 min at room temperature. The bicinchoninic acid assay (BCA assay) kit (Thermo Fisher Scientific) was used to measure protein concentration. Cell lysates were processed for SDS-PAGE and immunoblotting with standard procedure. All immunoblottings were developed by chemiluminescence using the SuperSignal West Dura reagents (Thermo Fisher Scientific). For SMase treatment assay, cells were cultured in DMEM supplemented with 10% lipoprotein-deficient serum (LPDS) (Sigma-Aldrich/Merck) and 50 µM mevastatin (Santa Cruz) for 16 hr, and then treated with 100 mU/ml SMase in the same culture media at 37°C for the indicated time before cell lysis (*Figure 6B–E* and *Figure 6—figure supplement 1A, B*). For experiments with recombinant EGFP–D4 proteins, cells that had been treated in the same way as the SMase treatment assay were washed once with PBS and subsequently incubated with PBS containing recombinant EGFP–D4 proteins (10 µg/ml) for 15 min at room temperature. Cells were then washed three times with PBS without EGFP–D4 proteins and immediately lysed (*Figure 6—figure supplement 2A,B*).

### Immunoprecipitation

HeLa cells expressing the indicated constructs were washed in cold PBS and lysed on ice in lysis buffer [50 mM Tris, 150 mM NaCl, 1% NP-40, 0.5 mM EDTA, 10% glycerol (pH 7.4) and protease inhibitor cocktail (Complete, mini, EDTA-free; Roche)]. Cell lysates were then centrifuged at 21,000 g for 20 min at 4°C. For anti-GFP and anti-Myc immunoprecipitation, supernatants were incubated with GFP-trap and Myc-trap agarose beads (Chromotek), respectively, for 30 min at 4°C under rotation. Subsequently, beads were washed in lysis buffer containing 1% NP-40 once and 0.2% NP-40 twice. Afterwards, immunoprecipitated proteins bound to the beads were incubated in PAGE sample loading buffer (containing 2% SDS) and then incubated at 60°C for 20 min and 70°C for 10 min. Immunoprecipitates were processed for SDS-PAGE and immunoblottings were carried out as described above.

## Protein purification

### Expression and purification of EGFP–D4, EGFP–D4H, GRAMD1c$_{StART}$, GRAM$_{1a}$ and GRAM$_{1b}$

All proteins were overexpressed in *E. coli* BL21-DE3 Rosetta cells. Inoculation cultures were started in 20 ml Terrific Broth (TB) medium supplemented with appropriate antibiotics. The cultures were incubated at 37°C, 200 rpm overnight. The following morning, bottles of 750 ml TB supplemented with appropriate antibiotics and 100 ul of antifoam 204 (Sigma-Aldrich/Merck) were inoculated with the inoculation cultures. The cultures were incubated at 37°C in the large-scale expression (LEX)

system with aeration and agitation through the bubbling of filtered air through the cultures. When the $OD_{600}$ reached ~2, the temperature was reduced to 18°C and the cultures were induced with 0.5 mM isopropyl β-d-1-thiogalactopyranoside (IPTG). Protein expression was allowed to continue overnight. The following morning, cells were harvested by centrifugation at 4,200 rpm, 15°C for 10 min and re-suspended in lysis buffer [100 mM HEPES, 500 mM NaCl, 10 mM imidazole, 10% glycerol, 0.5 mM TCEP(pH 8.0)], supplemented with proteinase inhibitors (Protease Inhibitor Cocktail Set III, EDTA free; Calbiochem) together with benzonase nuclease (Sigma-Aldrich/Merck)]. The re-suspended cell pellet suspensions were sonicated on ice on a Vibra Cell sonicator (Sonics and Materials, Inc) (70% power, 3 s pulse on, 3 s pulse off for 3 min). The lysate was clarified by centrifugation at 47,000 g, 4°C for 25 min. The supernatants were filtered through 1.2 µm syringe filters and loaded directly onto AKTA Xpress system (GE Healthcare). The lysates were loaded onto immobilized metal affinity chromatography (IMAC) columns. The sample-loaded columns were washed with 20 column volumes (CV) of wash buffer 1 [20 mM HEPES, 500 mM NaCl, 10 mM Imidazole, 10% glycerol, 0.5 mM TCEP (pH 7.5)] and 20 CV of wash buffer 2 [20 mM HEPES, 500 mM NaCl, 25 mM imidazole, 10% glycerol, 0.5 mM TCEP (pH 7.5)] or until a stable baseline was obtained. 5 CV of elution buffer 1 [20 mM HEPES, 500 mM NaCl, 500 mM imidazole, 10% glycerol, 0.5 mM TCEP (pH 7.5)] was used to elute proteins from IMAC columns and stored in sample loops on the system before being injected into gel filtration columns for further purification, using elution buffer 2 (20 mM HEPES, 300 mM NaCl, 10% glycerol, 0.5 mM TCEP, pH 7.5). Relevant peaks were pooled, and the protein sample was concentrated in Vivaspin 20 filter concentrators (VivaScience).

## Expression and purification of EGFP–helix, EGFP–helix (5E), GRAMD1a$_{StART}$, GRAMD1b$_{StART}$, GRAMD1b$_{StART}$ T469D and 5P mutants of GRAMD1a$_{StART}$ and GRAMD1b$_{StART}$

All proteins were overexpressed in *E. coli* BL21-DE3 Rosetta cells. The cell cultures were grown at 37°C until $OD_{600}$ reached ~0.5–0.7 with appropriate antibiotics. 0.1 mM IPTG (Thermo Fisher Scientific) was then added, and the culture was further grown at 18°C for 14–18 hr to allow protein expression. Cells were harvested by centrifugation at 4,700 g, at 4°C for 15 min, and re-suspended in lysis buffer [100 mM HEPES, 500 mM NaCl, 10 mM imidazole, 10% glycerol, 0.5 mM TCEP (pH 7.5)], supplemented with protease inhibitors (Complete, EDTA-free; Roche) together with benzonase nuclease (Sigma-Aldrich/Merck) or the cocktail of 100 ug/ml lysozyme (Sigma-Aldrich/Merck) and 50 ug/ml DNAse I (Sigma-Aldrich/Merck). Cells were lysed with sonication on ice in a Vibra Cell (Sonics and Materials, Inc) (70% power, 3 s pulse on, 3 s pulse off for 3 min for five rounds). The lysate was clarified by centrifugation at 47,000 g, at 4°C for 20 min. The supernatants were incubated at 4°C for 30 min with either 1 ml Co-TALON (Takara Bio Inc) or Ni-NTA resin (Thermo Fisher Scientific), which had been equilibrated with 10 ml of wash buffer 1. The protein-resin mixtures were then loaded onto a column to be allowed to drain by gravity. The column was washed with 10 ml of wash buffer 1 twice and 10 ml of wash buffer two twice, and then eluted with 5 ml of elution buffer 1. The proteins were then concentrated using Vivaspin 20 MWCO 10 kDa (GE Healthcare) or Amicon ultra-15 MWCO 10 kDa (Merck) and further purified by gel filtration (Superdex 200 increase 10/300 GL, GE Healthcare) with elution buffer 2, using the AKTA Pure system (GE Healthcare). Relevant peaks were pooled, and the protein sample was concentrated.

## Blue native (BN)-PAGE analysis of the EGFP-helix and EGFP-helix (5E)

BN-PAGE was performed using the Native Page Novex Bis-Tris Gel System (Thermo Fisher Scientific) according to the manufacturer's instructions. Briefly, 1 µg of purified proteins [EGFP-helix and EGFP-helix (5E)] were loaded on a 3–12% Bis-Tris gel. Electrophoresis was performed at 4°C at constant 150 V for 1 hr and then at constant 250 V. The gel was stained by colloidal blue (Thermo Fisher Scientific) according to the manufacturer's instruction. Gel filtration calibration kit LMW (GE healthcare) was used to provide marker proteins.

## Liposome-based experiments

### Liposome preparation

Lipids in chloroform were dried under a stream of $N_2$ gas, followed by further drying in the vacuum for 2 hr. Mole% of lipids used for the acceptor and donor liposomes in FRET-based lipid transfer

assays are shown in *Supplementary file 3*. The dried lipid films were hydrated with HK buffer [50 mM HEPES, 120 mM potassium acetate (pH 7.5)]. Liposomes were then formed by five freeze-thaw cycles (liquid $N_2$ and 37°C water bath) followed by extrusion using Nanosizer with a pore size of 100 nm (T and T Scientific Corporation). All liposomes except those shown in *Figure 3—figure supplement 1D* were subjected to extrusion.

### FRET-based DHE transfer assays

Buffer of the purified proteins was replaced with HK buffer prior to the FRET-based DHE transfer assay. Reactions were performed in 50 µl volumes. The final lipid concentration in the reaction was 1 mM, with donor and acceptor liposomes added at a 1:1 ratio (only acceptor liposomes contain 2.5% DNS-PE, see *Supplementary file 3* for the lipid compositions). Reactions were initiated by the addition of protein to a final concentration of 0.5–2 µM in a 96-well plate (Corning). The fluorescence intensity of DNS-PE (i.e. FRET signals), resulting from FRET between DNS-PE and DHE (excited at 310 nm), was monitored at 525 nm every 15 s over 30 min at room temperature by using a Synergy H1 microplate reader (Biotek). The values of blank solution (buffer only) were subtracted from all the values from each time point, and data were presented by setting the first value to zero at t = 0.

In some experiments, data were expressed as the number of DHE molecules transferred using the calibration curve (*Figure 5—figure supplement 1B–E*). For the generation of the calibration curve, FRET signals were measured for the liposomes containing 0%, 5% (1.25 nmole), 10% (2.5 nmole) or 15% (3.75 nmole) DHE and 2.5% DNS-PE (0.5 mM lipids in total: compositions of the liposomes can be found in *Supplementary file 3*). The mean of FRET signals at t = 0 from three replicates were plotted against the DHE mole number in liposomes (*Figure 5—figure supplement 1A*). Then, the mole number of the transferred DHE from the donor to acceptor liposomes in in vitro DHE transfer assay was obtained using the following formula: $y = 7649.6667 + 5946.61531x$ (derived from the linear fit of the calibration curve). To obtain $x$ (the amount of transferred DHE in nmole shown in the y axis of *Figure 5—figure supplement 1F*), the FRET values from each time point of the in vitro lipid transfer assay were substituted for the $y$ of the equation. Transfer rates of individual StART-like domain were obtained from the slopes of the graphs using the one-phase association function of Prism 7 (GraphPad) (*Figure 5—figure supplement 1F*).

### Liposome sedimentation assays

Heavy liposomes were prepared by hydrating 1.6 mM dried lipid films in HK buffer containing sucrose [50 mM HEPES (pH 7.5), 120 mM K-acetate and 0.75 M sucrose] and subjected to freeze-thaw cycles five times. Next, 200 µl of heavy liposomes were pelleted and washed with HK buffer without sucrose twice to remove unencapsulated sucrose. Pelleted heavy liposomes were resuspended in 200 µl HK buffer and incubated with 7.5–10 µg of the indicated proteins for 1 hr at room temperature. Unbound proteins (supernatant) were separated from liposome-bound proteins (pellet) by centrifugation at 21,000 x g for 1 hr at 25°C. After centrifugation, the supernatant was removed, and pellets were re-suspended in 200 µl HK buffer. 20 µl samples were taken from both fractions and run on SDS-PAGE followed by colloidal blue staining. Quantification of the bands was performed using Fiji.

## Molecular modeling

The modeled structure of GRAMD1b$_{StART}$ was obtained by submitting the primary sequence (residues 375–545) to the I-TASSER server, using the GRAMD1a$_{StART}$ structure (PDB: 6GQF) as the template.

Primary sequences of luminal helices of GRAMD1s (GRAMD1a, 657–706; GRAMD1b, 672–720; GRAMD1c, 599–647) were submitted to the I-TASSER server without assigning any templates. Only the luminal amphipathic helix region, indicated in *Figure 2A*, is shown in *Figure 2—figure supplement 1B–C*.

## Statistical analysis

No statistical method was used to predetermine sample size, and the experiments were not randomized for live-cell imaging. Sample size and information about replicates are described in the figure legends. The number of biological replicates for all cell-based experiments and the number of

technical replicates for all other biochemical assays are shown as the number of independent experiments within the figure legends for each figure. Comparisons of data were carried out by the two-tailed unpaired Student's t-test, Holm-Sidak's t-test or one-way ANOVA, followed by Tukey or Dunnett corrections for multiple comparisons as appropriate with Prism 7 or 8 (GraphPad software). Unless $p < 0.0001$, exact P values are shown within the figure legends for each figure. $p > 0.05$ was considered not significant.

## Acknowledgements

We thank Pietro De Camilli, Min Wu, Luo Dahai, Darshini Jeyasimman, Jingbo Sun, Nur Raihanah Binte Mohd Harion and Dhakshenya Dhinagaran for discussion and/or sharing reagents. We thank the NTU Protein Production Platform (www.proteins.sg) for the cloning, expression tests and purification of the wild-type StART-like domains of GRAMD1a, GRAMD1b, GRAMD1c, the GRAM domains of GRAMD1a and GRAMD1b, and the recombinant EGFP–D4 and EGFP–D4H proteins. This work was supported in part by the Singapore Ministry of Education Academic Research Fund Tier 2 (MOE2017-T2-2-001), a Nanyang Assistant Professorship (NAP), a Lee Kong Chian School of Medicine startup grant (LKCMedicine-SUG), and a Grant-in-Aid for Young Scientists (A) from the Japan Society for the Promotion of Science (17H05065) to YS. MRW, FTT and AT were supported by grants from the National University of Singapore via the Life Sciences Institute (LSI) and the National Research Foundation (NRFI2015-05 and NRFSBP-P4). TN was supported by a fellowship from the Japanese Society for Promotion of Science.

## Additional information

### Funding

| Funder | Grant reference number | Author |
|---|---|---|
| Japan Society for the Promotion of Science | 17H05065 | Yasunori Saheki |
| Ministry of Education - Singapore | MOE2017-T2-2-001 | Yasunori Saheki |
| Nanyang Technological University | Nanyang Assistant Professorship (NAP) | Yasunori Saheki |
| Nanyang Technological University | Lee Kong Chian School of Medicine startup grant | Yasunori Saheki |
| National University of Singapore | Life Sciences Institute NRFI2015-05 | Alexander Triebl Federico Tesio Torta Markus R Wenk |
| National Research Foundation Singapore | NRFSBP-P4 | Alexander Triebl Federico Tesio Torta Markus R Wenk |
| Japan Society for the Promotion of Science | Overseas Research Fellowship | Tomoki Naito |

The funders had no role in study design, data collection and interpretation, or the decision to submit the work for publication.

### Author contributions

Tomoki Naito, Conceptualization, Data curation, Formal analysis, Validation, Investigation, Methodology, Writing—review and editing, Designed and performed all aspects of genetic manipulations, Designed and performed all aspects of imaging and biochemical studies, Performed the isolation of plasma membrane sheets; Bilge Ercan, Conceptualization, Data curation, Formal analysis, Validation, Investigation, Methodology, Writing—review and editing, Designed and performed all aspects of lipid transfer assays, Designed and performed all aspects of liposome sedimentation assays, Performed all the structural modeling and designed StART-like domain mutations, Designed and performed protein purification work; Logesvaran Krshnan, Conceptualization, Validation, Investigation,

Methodology, Designed and performed genetic manipulations, Designed imaging and biochemical studies; Alexander Triebl, Formal analysis, Investigation, Methodology, Performed lipidomics analysis; Dylan Hong Zheng Koh, Resources, Data curation, Formal analysis, Validation, Investigation, Methodology, Writing—review and editing, Designed and performed genetic manipulations, Designed and performed imaging and biochemical studies, Performed the isolation of plasma membrane sheets; Fan-Yan Wei, Resources, Writing—review and editing; Kazuhito Tomizawa, Resources, Supervision; Federico Tesio Torta, Formal analysis, Supervision, Investigation, Methodology, Performed lipidomics analysis; Markus R Wenk, Resources, Supervision, Funding acquisition, Methodology, Performed lipidomics analysis; Yasunori Saheki, Conceptualization, Resources, Data curation, Formal analysis, Supervision, Funding acquisition, Validation, Investigation, Visualization, Methodology, Writing—original draft, Project administration, Writing—review and editing, Designed all experiments, Supervised the project, Performed the isolation of plasma membrane sheets

### Author ORCIDs
Alexander Triebl (iD) https://orcid.org/0000-0001-8423-8224
Yasunori Saheki (iD) https://orcid.org/0000-0002-1229-6668

### Decision letter and Author response
Decision letter https://doi.org/10.7554/eLife.51401.sa1
Author response https://doi.org/10.7554/eLife.51401.sa2

## Additional files

### Supplementary files
• Supplementary file 1. Key resources table.

• Supplementary file 2. Table 1. A list of sequence-based reagents. DNA sequences for oligos and primers used in this study are described.

• Supplementary file 3. Table 2. Lipid compositions of liposomes used for lipid transfer assays. Moles % of lipids used for the acceptor and donor liposomes in FRET-based lipid transfer experiments are described.

• Transparent reporting form

### Data availability
All data generated or analyzed during this study are included in the manuscript and supporting files. Source data files have been provided for Figures 2, 3, 4, 5, 6, 7, 3-S-1, 3-S-2, 4-S-2, 4-S-3, 5-S-1, 5-S-2, 6-S-1, 6-S-2, 7-S-1, and 7-S-2.

The following previously published dataset was used:

| Author(s) | Year | Dataset title | Dataset URL | Database and Identifier |
|---|---|---|---|---|
| Fairall L, Gurnett JE, Vashi D, Sandhu J, Tontonoz P, Schwabe JWR | 2018 | The structure of mouse AsterA (GramD1a) with 25-hydroxy cholesterol | https://www.rcsb.org/structure/6GQF | Protein Data Bank, 6GQF |

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
