## [Decision Letter]

**Acceptance summary:**

Cells go to great lengths to maintain optimal levels of cholesterol in their membranes. Most of a cell's cholesterol resides in its plasma membrane (PM), however the regulatory machinery that ensures cholesterol homeostasis is located in the endoplasmic reticulum (ER) membrane. How the PM communicates its cholesterol status to the ER is still not well understood, although the distribution of PM cholesterol into accessible and inaccessible pools has been shown to play a role. Here, Naito et al. show that the ER-localized GRAMD1 proteins form oligomers, accumulate at ER-PM contact sites, and bind and transport accessible PM cholesterol to the ER. These findings are consistent with and build upon earlier work on these proteins in yeast and mammalian cells. Cells lacking all three GRAMD1 proteins are unable to relieve the accumulation of accessible cholesterol in the PM. This study brings a deeper understanding of the players involved in intracellular cholesterol transport and will thus be of broad interest to scientists in the fields of cell biology and lipid metabolism.

**Decision letter after peer review:**

Thank you for submitting your article "The regulation of plasma membrane cholesterol states by GRAMD1 lipid transfer protein complex" for consideration by *eLife*. Your article has been reviewed by three peer reviewers, and the evaluation has been overseen by a Reviewing Editor and Suzanne Pfeffer as the Senior Editor. The reviewers have opted to remain anonymous.

The reviewers have discussed the reviews with one another and the Reviewing Editor has drafted this decision to help you prepare a revised submission.

Summary:

The manuscript is well-written and makes important contributions to the field of intracellular cholesterol trafficking. The authors have added to the field by showing that GRAMD1 proteins may affect the distribution of cholesterol in plasma membranes (PMs) between accessible and inaccessible forms. Studies that support these points include a careful analysis of the cholesterol-sensing properties of GRAM domains and phenotypes of complete (triple) GRAMD1 knockout cells. Other parts of these studies showing the enrichment of GRAMD1s at ER-PM contact sites, binding/transport of cholesterol in vitro, and facilitation of PM-to-ER cholesterol transport nicely confirm previous conclusions from work on these proteins and their yeast homologues. All the reviewers were enthusiastic about this paper, but also raised the following concerns that need to be addressed before acceptance.

Essential revisions:

1) A key point of this paper is that the GRAMD1 proteins sense and influence levels of accessible cholesterol in PMs. One nice piece of evidence provided by the authors is that accessible cholesterol generated after SMase treatment is not transported to ER to block SREBP activation in GRAMD1 triple knockout cells. However, more direct measurements of changes in accessible cholesterol would be desirable. The experiments where fluorescent D4H is expressed in cells cannot be interpreted because of the severe perturbing effects of the D4 probes in live cells. The authors do point out this limitation and we recommend removing these studies. Instead, endpoint assays where direct binding of D4H or ALOD4 or other toxin variants is measured after various treatments would be more informative and should be performed. Another option is to measure the decline in sphingomyelin-sequestered or inaccessible cholesterol using the OlyA probe (Endapally et al., 2019).

2) If GRAMD1 proteins play a unique role in determining the size of accessible cholesterol pools in the PM, then their effect should not be influenced by high-level expression of other cholesterol transporters such as STARD4 or the OSBP family members. This can be easily tested by the authors in the cell lines they have generated.

3) The liposome experiments showing the interaction of GRAM domains with cholesterol-containing membranes are well executed. To rule out the possibility of other membrane factors affecting binding, the authors could test whether other non-bilayer forming lipids such as PE can replace cholesterol. Also, the phospholipid acyl chain saturation has profound effects on accessibility of cholesterol. This can also be tested.

4) References to previous contributions to this field are incomplete. The authors reference several original papers, which we commend them on, but they also unnecessarily reference additional reviews in many cases. The referencing is also repetitive, since the same block of references are cited at multiple points. It would add, not detract, from the impact of this paper if the authors were to be generous to those who made earlier contributions. Specific cases are:

a) While it is true that the Tontonoz paper focused on the role of GRAMD1 proteins in handling HDL-derived cholesterol in the adrenal gland, it is unfair to say that their paper failed to appreciate a more global role for Asters/GRAMs in intracellular cholesterol trafficking. Indeed, a more general role in sensing accessible cholesterol was explicitly proposed in the first paragraph of the Discussion of that paper. This should be acknowledged here.

b) The authors should mention in the Discussion section the intriguing possibility that some cell types, such as macrophages, can dispose of accessible cholesterol by release of PM particles (aside from non-vesicular transport to ER).

c) The authors neglect the seminal contributions of the McConnell group at Stanford and others, which inspired the later studies. These earlier papers first introduced the concept of complexes, chemical activity, and "accessibility" and should be referenced at appropriate points in the paper (Radhakrishnan et al., 2000, McConnell and Radhakrishnan, 2003, Sokolov and Radhakrishnan, 2010, Gay et al., 2015).

5) Please include a statement in the Introduction to clarify the nomenclature difference between Asters and GRAMD1s.

[Editors' note: further revisions were requested prior to acceptance, as described below.]

Thank you for submitting your article "The regulation of plasma membrane cholesterol states by GRAMD1 lipid transfer protein complex" for consideration by *eLife*. Your revised article has been evaluated by a Reviewing Editor and Suzanne Pfeffer as the Senior Editor. The reviewers have discussed the reviews with one another and the Reviewing Editor has drafted this decision to help you prepare a revised submission.

Summary:

The revised submission by Naito et al. has improved and has addressed some of the reviewer concerns. The new experiments in Figure 3 and its supplements are convincing in showing the specificity of the GRAM domains for accessible cholesterol. However, a few important points that were raised by the reviewers remain unresolved and need additional attention.

Essential revisions:

The main concern centers on the authors' claim that GRAMD1 proteins not only transport cholesterol between the plasma membrane (PM) and the ER (which has already been shown by the Tontonoz group and by papers on the yeast homologs), but that they also affect the distribution of cholesterol between accessible and inaccessible pools in the PM. The data supporting this latter claim (including the new Figure 6—figure supplement 2) are not very convincing. If GRAMD1 proteins are increasing the pool of accessible cholesterol, but not changing total cholesterol, then inaccessible cholesterol must be decreasing. How do the authors propose that the GRAMD1 proteins accomplish this redistribution? Probing for such a decrease using probes for inaccessible cholesterol (Endapally et al., 2019) would strengthen this argument. Alternatively, the data can be explained simply by a role for GRAMD1s in transporting accessible cholesterol, but not in regulating the accessibility of PM cholesterol.

The authors seem to have misunderstood the suggestion to test whether STARD4 or ORPs can substitute for GRAMD1 proteins. The question is not whether these other transporter proteins can localize to the PM but whether they can also influence the size of the accessible cholesterol pool (by D4H binding for instance). This measurement may give insights into unique roles of GRAMD1 proteins.

These points need to be addressed to give some mechanistic insights into the functioning of GRAMD1 proteins.

---

## [Author Response]

Essential revisions:1) A key point of this paper is that the GRAMD1 proteins sense and influence levels of accessible cholesterol in PMs. One nice piece of evidence provided by the authors is that accessible cholesterol generated after SMase treatment is not transported to ER to block SREBP activation in GRAMD1 triple knockout cells. However, more direct measurements of changes in accessible cholesterol would be desirable. The experiments where fluorescent D4H is expressed in cells cannot be interpreted because of the severe perturbing effects of the D4 probes in live cells. The authors do point out this limitation and we recommend removing these studies. Instead, endpoint assays where direct binding of D4H or ALOD4 or other toxin variants is measured after various treatments would be more informative and should be performed. Another option is to measure the decline in sphingomyelin-sequestered or inaccessible cholesterol using the OlyA probe (Endapally et al., 2019).

We agree with the importance of more direct measurements of changes in accessible cholesterol. We appreciate the constructive suggestions and addressed the concerns by performing additional experiments.

First, we monitored changes in accessible PM cholesterol after SMase treatment more directly, using a recombinant EGFP-tagged D4 of PFO protein (EGFP-D4) via endpoint assays. In brief, wild-type control and GRAMD1 TKO cells were treated with SMase for a fixed period of time (0 min, 30 min, 60 min, 90 min, 120 min, 150 min, 180 min), washed, and then incubated with recombinant EGFP-D4 proteins. After wash, cell lysates were collected and analyzed by SDS-PAGE followed by immuno-blotting against GFP to detect EGFP-D4 proteins that were bound to accessible cholesterol in the PM. 30 min treatment with SMase induced similar extent of the binding of EGFP-D4 to both control and TKO cells. Gradual decrease of EGFP-D4 signals was observed in control cell lysates over the time course of 180 min. TKO cells, however, showed persistent increase in binding of EGFP-D4 to the PM even after 180 min. These new results suggest that acute extraction of accessible PM cholesterol is able to counteract with acute expansion of the accessible pool of PM cholesterol to prevent its further expansion in wild-type control cells and such homeostatic response is impaired in GRAMD1 TKO cells, resulting in larger expansion of the accessible pool of PM cholesterol in these cells. These results are now included in new Figure 6—figure supplement 2with relevant text changes in the fourth paragraph of the subsection “GRAMD1s play a role in PM to ER cholesterol transport during acute expansion of the accessible pool of PM cholesterol”

Second, we removed all the data sets where D4H was expressed in cells from our revised manuscript (old Figure 7, Figure 7—figure supplement 1, Figure 7—figure supplement 2, Figure 7—figure supplement 3, and Figure 7—figure supplement 4)and replaced them with new data sets that we obtained from microscopy-based endpoint assays where a recombinant EGFP-D4H protein was used as a probe to assess the accessible pool of PM cholesterol. We repeated various rescue experiments, including the rapamycin-induced acute PM recruitment of chimeric GRAMD1b proteins, and obtained results that confirm the roles of GRAMD1s in regulating the size of accessible cholesterol pools in the PM. These new results are now included in new Figure 7, new Figure 7—figure supplement 1, new Figure 7—figure supplement 2, and new Figure 7—figure supplement 3. Accordingly, we made necessary text changes in the subsections “PM cholesterol states are altered in GRAMD1 TKO cells” and “Acute recruitment of GRAMD1b to ER-PM contacts in GRAMD1 TKO cells restores PM cholesterol states”. Our results are consistent with the important roles of GRAMD1s in regulating the accessibility of cholesterol in the PM.

2) If GRAMD1 proteins play a unique role in determining the size of accessible cholesterol pools in the PM, then their effect should not be influenced by high-level expression of other cholesterol transporters such as STARD4 or the OSBP family members. This can be easily tested by the authors in the cell lines they have generated.

Thank you for suggesting this important experiment. In the previous manuscript, we showed that SMase treatment induced enhanced PM recruitment of exogenously expressed EGFP-tagged GRAM domain of GRAMD1b (EGFP-GRAM_1b_) in GRAMD1 TKO cells and that such phenotype was rescued by re-expression of GRAMD1b. Using this assay, we tested if overexpression of other cholesterol transporters, including STARD4 and selected OSBP family members could compensate the loss of GRAMD1s (thereby rescuing the enhanced PM recruitment of EGFP-GRAM_1b_ in GRAMD1 TKO cells). We expressed either one of mCherry-tagged STARD4, mRuby-tagged OSBP, mRuby-tagged ORP4, or mRuby-tagged ORP9, individually together with EGFP-GRAM_1b_ in GRAMD1 TKO cells and stimulated cells with SMase. While GRAMD1b was rapidly recruited to the PM upon SMase treatment, none of the other cholesterol transporters were recruited to the PM, demonstrating the unique property of GRAMD1s to respond to acute increase in the PM cholesterol accessibility. Accordingly, none of these proteins suppressed the enhanced PM recruitment of EGFP-GRAM_1b_ in TKO cells. These new results are now included in Figure 5—figure supplement 2 with relevant text changes in the subsection “The cholesterol transporting property of the StART-like domain of GRAMD1s is critical for the clearance of the accessible pool of PM cholesterol”.

3) The liposome experiments showing the interaction of GRAM domains with cholesterol-containing membranes are well executed. To rule out the possibility of other membrane factors affecting binding, the authors could test whether other non-bilayer forming lipids such as PE can replace cholesterol. Also, the phospholipid acyl chain saturation has profound effects on accessibility of cholesterol. This can also be tested.

Thank you for the nice comments on our liposome-based experiments. We performed additional liposome experiments as suggested by the reviewers and included these new results in the revised manuscript.

First, we examined if PE would show similar effects as cholesterol by replacing cholesterol with PE. The GRAM domain did not bind liposomes that contained PE instead of cholesterol, demonstrating the specificity of cholesterol in the binding of the GRAM domain. These new results are included in new Figure 3—figure supplement 2A with relevant text changes in the subsection “The GRAM domains of GRAMD1s act as coincidence detector of unsequestered cholesterol and anionic lipids and sense the accessibility of cholesterol”.

Second, we examined the effect of the phospholipid acyl chain saturation and branching on membrane binding of the GRAM domain. We generated liposomes containing fixed amounts of phosphatidylserine (20%) with varying ratios of cholesterol and phosphatidylcholine. We individually tested either one of the three types of phosphatidylcholine that possesses different acyl chain structures, namely POPC, DOPC, and DPhyPC. The binding of GRAM domain of GRAMD1b to liposomes shifted to lower cholesterol concentration as the ordering tendency of phosphatidylcholine is lowered (i.e. as cholesterol sequestration effect is reduced). These results are consistent with the ability of the GRAM domain to sense the accessibility of cholesterol in membranes. Citing several key references suggested by the reviewers (e.g. Radhakrishnan et al., 2000, Sokolov and Radhakrishnan, 2010, Gay et al., 2015), these new results are now included in new Figure 3—figure supplement 2B, C with relevant text changes in the sixth paragraph of the subsection “The GRAM domains of GRAMD1s act as coincidence detector of unsequestered cholesterol and anionic lipids and sense the accessibility of cholesterol”.

4) References to previous contributions to this field are incomplete. The authors reference several original papers, which we commend them on, but they also unnecessarily reference additional reviews in many cases. The referencing is also repetitive, since the same block of references are cited at multiple points. It would add, not detract, from the impact of this paper if the authors were to be generous to those who made earlier contributions. Specific cases are:

We apologize for the incomplete referencing to previous contributions to the filed. In the revised version of the manuscript, we did our best to cite original papers wherever it is appropriate and reduced the number of unnecessary reviews. We also avoided citing the same block of references.

a) While it is true that the Tontonoz paper focused on the role of GRAMD1 proteins in handling HDL-derived cholesterol in the adrenal gland, it is unfair to say that their paper failed to appreciate a more global role for Asters/GRAMs in intracellular cholesterol trafficking. Indeed, a more general role in sensing accessible cholesterol was explicitly proposed in the first paragraph of the Discussion of that paper. This should be acknowledged here.

We agree with the reviewers’ comment and apologize for this mistake. We toned down our sentences and acknowledged the contribution of the Tontonoz paper more properly. We also added the following sentence in the Discussion section: “Based on these results, they proposed that GRAMD1s might also play more general roles in intracellular cholesterol transport by sensing accessible cholesterol in the PM.”

b) The authors should mention in the Discussion section the intriguing possibility that some cell types, such as macrophages, can dispose of accessible cholesterol by release of PM particles (aside from non-vesicular transport to ER).

Thank you for bringing this exciting possibility to our attention. We agree with the reviewers that such release of PM particles and GRAMD1s-mediated PM to ER transport can act as parallel mechanisms to remove accessible PM cholesterol. We cited the recent papers by the Young lab (Hu et al., 2019; He et al., 2018) in the Discussion section.

c) The authors neglect the seminal contributions of the McConnell group at Stanford and others, which inspired the later studies. These earlier papers first introduced the concept of complexes, chemical activity, and "accessibility" and should be referenced at appropriate points in the paper (Radhakrishnan et al., 2000, McConnell and Radhakrishnan, 2003, Sokolov and Radhakrishnan, 2010, Gay et al., 2015).

We apologize for omitting the references to the seminal contribution by the McConnell lab and the Radhakrishnan Lab. We cited all the four papers at appropriate points in the revised manuscript (e.g. Introduction, subsection “The GRAM domains of GRAMD1s act as coincidence detector of unsequestered cholesterol and anionic lipids and sense the accessibility of cholesterol”, and subsection “GRAMD1s play a role in PM to ER cholesterol transport during acute expansion of the accessible pool of PM cholesterol”).

5) Please include a statement in the Introduction to clarify the nomenclature difference between Asters and GRAMD1s.

We added the following statement in the Introduction to clarify the nomenclature: “These GRAMDs include the StART-like domain-containing GRAMD1s, also known as Asters (GRAMD1a/Aster-A, GRAMD1b/Aster-B, and GRAMD1c/Aster-C)…”

[Editors' note: further revisions were requested prior to acceptance, as described below.]

Essential revisions:The main concern centers on the authors' claim that GRAMD1 proteins not only transport cholesterol between the plasma membrane (PM) and the ER (which has already been shown by the Tontonoz group and by papers on the yeast homologs), but that they also affect the distribution of cholesterol between accessible and inaccessible pools in the PM. The data supporting this latter claim (including the new Figure 6—figure supplement 2) are not very convincing. If GRAMD1 proteins are increasing the pool of accessible cholesterol, but not changing total cholesterol, then inaccessible cholesterol must be decreasing. How do the authors propose that the GRAMD1 proteins accomplish this redistribution? Probing for such a decrease using probes for inaccessible cholesterol (Endapally et al., 2019) would strengthen this argument. Alternatively, the data can be explained simply by a role for GRAMD1s in transporting accessible cholesterol, but not in regulating the accessibility of PM cholesterol.

Thank you for raising this important concern. We would like to clarify our claim. First of all, we *did not* propose that GRAMD1 proteins are increasing the pool of accessible cholesterol in any part of our original/revised manuscript. Instead, we have been proposing that the key function of GRAMD1s is to monitor/sense increase in accessible PM cholesterol by their GRAM domains and to transport accessible PM cholesterol to the ER by their StART-like domain when the accessible pool of PM cholesterol transiently expands. Such homeostatic function is important in cells to maintain normal levels of accessible cholesterol in the PM (i.e. to limit accessibility of PM cholesterol). This is evidenced by our original and new data showing that the lack of GRAMD1s results in changes in accessible pool of cholesterol in the PM in acute (e.g. sustained accumulation upon SMase treatment: Figure 4 and Figure 6—figure supplement 2 and the related data within the section of “GRAMD1s play a role in PM to ER cholesterol transport during acute expansion of the accessible pool of PM cholesterol”) and chronic conditions (e.g. expansion at steady state: Figure 7 and Figure 7—figure supplement 2, using recombinant D4H proteins as specified by the reviewer in the first revision).

We agree with the reviewer/s that our data can be explained by a role for GRAMD1s in transporting accessible cholesterol. In fact, in the original Discussion section, we specifically stated, “All these phenotypes are consistent with defects in efficient PM to ER transport of the accessible pool of cholesterol.” We had also stated at the end of the Introduction, “Collectively, our findings provide evidence for novel cellular mechanisms by which GRAMD1s monitor and regulate PM cholesterol states in mammalian cells as one of the key homeostatic regulators. GRAMD1s dynamically localize to ER-PM contact sites and limit the accessibility of PM cholesterol by their ability to sense transient expansion of the accessible pool of PM cholesterol, and to facilitate its transport to the ER, thereby regulating PM cholesterol homeostasis.”

Furthermore, we showed in the new Figure 7 (specifically by using endpoint assays where we measured direct binding of D4H to the PM after various treatments as suggested by the reviewer in the first revision) that the forced recruitment of wild-type GRAMD1b (but not a mutant version that is defective in cholesterol transport) to ER-PM contacts can reduce the binding of recombinant EGFP-D4H proteins to the PM in GRAMD1 TKO cells (Figure 7E, F). These results further support the direct role of GRAMD1s in extracting and transporting accessible cholesterol from the PM to the ER at these contacts (thereby limiting the accessibility of PM cholesterol).

Although we tried to explain this notion (i.e. our data can be explained by a role for GRAMD1s in transporting accessible cholesterol) explicitly in our original and revised manuscript, we realize that we use “regulating the accessibility of PM cholesterol” in some parts of our manuscript as if it was another separable function of GRAMD1s (including within the previous rebuttal letter). We apologize if such phrasing caused confusion regarding the functions of GRAMD1s to the reviewer/s. We have revised the texts throughout the manuscript to remove or clarify “regulating the accessibility of PM cholesterol” to be consistent with our claim.

The authors seem to have misunderstood the suggestion to test whether STARD4 or ORPs can substitute for GRAMD1 proteins. The question is not whether these other transporter proteins can localize to the PM but whether they can also influence the size of the accessible cholesterol pool (by D4H binding for instance). This measurement may give insights into unique roles of GRAMD1 proteins.

We would like to clarify our experiments. We performed these experiments specifically to test if overexpression of STARD4 and ORPs would rescue/influence exaggerated expansion of the accessible pool of PM cholesterol in GRAMD1 TKO cells upon SMase treatment [by using EGFP-tagged GRAM domain (EGFP-GRAM_1b_) as a biosensor to detect increase in the size of accessible PM cholesterol pool in combination with total internal reflection fluorescence (TIRF) imaging technique to focus on the PM] (Figure 5—figure supplement 2B, D).

In the original manuscript, we presented that cells lacking GRAMD1s showed enhanced PM recruitment of EGFP-GRAM_1b_ upon SMase treatment compared to wild-type control cells (i.e. exaggerated expansion of the accessible pool of PM cholesterol) (Figure 4D, E). We showed that such enhanced PM recruitment of EGFP-GRAM_1b_ can be rescued by wild-type GRAMD1b but not by a mutant GRAMD1b that is defective in cholesterol transport (Figure 5E). Thus, our live cell imaging-based assay is sensitive enough to examine whether overexpression of other cholesterol transporters, including ORPs and STARDs, are sufficient to substitute the functions of GRAMD1s or not. Our results demonstrate that the function of GRAMD1s cannot be substituted by STARD4 or ORPs in our imaging-based assay (Figure 5—figure supplement 2B, D). Based on these considerations, we revised the text accordingly to avoid confusion (subsection “The cholesterol transporting property of the StART-like domain of GRAMD1s is critical for the clearance of the accessible pool of PM cholesterol”).